# Sobolev Gradient Ascent for Optimal Transport: Barycenter Optimization and Convergence Analysis

**Kaheon Kim**[*]
Department of ACMS
University of Notre Dame
Notre Dame, IN 46556
kkim26@nd.edu

**Bohan Zhou**[*]
Department of Mathematics
University of California
Santa Barbara, CA 93106
bhzhou@ucsb.edu

**Changbo Zhu**
Department of ACMS
University of Notre Dame
Notre Dame, IN 46556
czhu4@nd.edu

**Xiaohui Chen**
Department of Mathematics
University of Southern California
Los Angeles, CA 90089
xiaohuic@usc.edu

## Abstract

This paper introduces a new constraint-free concave dual formulation for the Wasserstein barycenter. Tailoring the vanilla dual gradient ascent algorithm to the Sobolev geometry, we derive a scalable Sobolev gradient ascent (SGA) algorithm to compute the barycenter for input distributions discretized over a regular grid. Despite the algorithmic simplicity, we provide a global convergence analysis that achieves the same rate as the classical subgradient descent methods for minimizing nonsmooth convex functions in the Euclidean space. A central feature of our SGA algorithm is that the computationally expensive $c$-concavity projection operator enforced on the Kantorovich dual potentials is unnecessary to guarantee convergence, leading to significant algorithmic and theoretical simplifications over all existing primal and dual methods for computing the exact barycenter. Our numerical experiments demonstrate the superior empirical performance of SGA over the existing optimal transport barycenter solvers.

## 1 Introduction

Recent years have seen an increasing trend of interest in applying the theory of optimal transport (OT) in statistics and machine learning. In these data-driven fields, a fundamental question is *how to average*? Averaging is useful to find common patterns in the input data and to summarize the algorithm output from multiple data sources. Unlike the tabular data, where each row is a data point with features being viewed as a vector in an Euclidean space, information in data with complex and often geometric structures (such as 2D images and 3D shapes) cannot be effectively extracted with conventional techniques. A pervasive OT analog of the mean for a collection of distributional input data $\mu_1, \ldots, \mu_n$ as probability measures on $\mathbb{R}^d$ is the Wasserstein barycenter introduced by Agueh & Carlier (2011), which is defined as any minimizer of the following barycenter functional

$$\mathcal{B}(\nu) \triangleq \sum_{i=1}^{m} \frac{\alpha_i}{2} W_2^2(\mu_i, \nu), \tag{1}$$

where $\alpha_i$'s are positive weights such that $\sum_{i=1}^{m} \alpha_i = 1$ and $W_2(\mu, \nu)$ is the 2-Wasserstein distance between two probability distributions $\mu$ and $\nu$. This variational formulation of "middle point" in Problem (1) is a natural extension of the variance functional for Euclidean random variables, and it is

---

[*]joint first author

applicable to general metric spaces. As in the conventional setting, the Wasserstein barycenter has been used in several machine learning tasks such as modeling the centroids in clustering measure-valued data (Zhuang et al., 2022; Ho et al., 2017) and aggregating distributional outputs from subsets of a massive dataset in Bayesian methods (Srivastava et al., 2018).

It remains a major challenge to compute the Wasserstein barycenter in a tractable (or even scalable) way with principled theoretical guarantees in the literature. For high-dimensional distributions, computing or even approximating the barycenter is worst-case NP-hard (Altschuler & Boix-Adsera, 2022). In light of the curse of dimensionality obstacle, we focus on computing the barycenter for fixed dimension $d$ in this paper. There are various primal-dual reformulations of the barycenter functional in (1) that lead to different optimization methods (Álvarez-Esteban et al., 2016; Zemel & Panaretos, 2019; Zhou & Parno, 2024; Kim et al., 2025) (cf. **Literature review** in Section 1.1).

## 1.1 LITERATURE REVIEW

Barycenter computation can be broadly categorized into two types, depending on whether the optimization problem (1) is solved in the primal space or dual space. Carlier, Guillaume et al. (2015); Ge et al. (2019) noted that the Wasserstein barycenter problem can be recast as a linear program (LP), which, despite being tractable, has poor scalability for large-scale problems (e.g., high-resolution grids or dense point clouds). Álvarez-Esteban et al. (2016) proposed a method directly solving the fixed point iterations on the primal variable in (1), which empirically works for the location-scatter family of distributions without quantitative convergence guarantee. First-order methods such as the Wasserstein gradient descent were considered in (Zemel & Panaretos, 2019; Chewi et al., 2020). In each iteration, one needs to compute $m$ OT maps, a highly undesirable feature for problems with a large (or even moderate) number of marginal distributions. Quantitative convergence rate is only available for Bures-Wasserstein barycenter, when all marginals are Gaussian distributions (Chewi et al., 2020).

On the other hand, a dual approach based on the multimarginal OT (MOT), originated from (Gangbo & Święch, 1998), was proposed by Zhou & Parno (2024), where the MOT problem was represented by a fully connected undirected graph. This approach avoids the computation of OT maps as in the primal problem, however with the cost of introducing $O(m^2)$ constraints in the dual problem, thus resulting an algorithm with per-iteration time complexity of $O(m^2 \times n \log n)$ for $m$ input distributions observed on a grid of size $n$. Moreover, there is no convergence guarantee established for this approach. More recently, Kim et al. (2025) proposed a nonconvex-concave minimax formulation that leverages the benefits from both the primal and dual structures. The resulting Wasserstein descent $\dot{\mathbb{H}}^1$-ascent (WDHA) algorithm achieves a convergence rate of $O(T^{-1/2})$ for $T$ iterations under a fairly expensive convex and smooth projection operation with a quadratic time complexity $O(n^2)$ in grid size $n$ per iteration. In practice, the projection is replaced with a conjugate envelope operation, which reduces the per-iteration complexity to $O(n \log n)$. Nonetheless, this substitution introduces a gap between the theoretically analyzable algorithm and the one that is computationally feasible in practice.

It is worth mentioning that there is an extensive literature on regularized notions and algorithms for OT barycenters (Cuturi & Doucet, 2014; Bigot et al., 2019; Janati et al., 2020; Li et al., 2020; Carlier et al., 2021; Fan et al., 2022; Chizat, 2023; Chi et al., 2023; Noble et al., 2023; Kolesov et al., 2024b; Li & Chen, 2025). Nevertheless, we emphasize that the current paper is mainly concerned with the computation of the *exact* Wasserstein barycenter without any regularization. High-accuracy Wasserstein barycenter has seen applications in, for example, medical imaging (Gramfort et al., 2015) and scalable Bayesian inference (Srivastava et al., 2018).

## 1.2 CONTRIBUTION AND LIMITATION

The goal of this paper is to compute the exact barycenter for a collection of input distributions with densities discretized over a regular grid. We propose a novel *constraint-free* and *concave* formulation of the Wasserstein barycenter optimization problem that achieves strong duality. Our barycenter formulation fully operates in the dual space, thus avoiding the OT map computations in the primal problem. This unconstrained concave structure allows us to derive a computationally simple and efficient gradient method in an appropriate dual geometry. Remarkably, we show that, without the computationally expensive $c$-concave projection steps (or equivalently, the convex conjugate

projection) in existing algorithms, our proposed Sobolev gradient ascent (SGA) algorithm retains a strong algorithmic convergence guarantee. In particular, we prove a global convergence rate that matches the rate of classical subgradient descent methods for minimizing nonsmooth convex functions in the Euclidean space (Nesterov, 2013; Lan, 2020). We report via numerical studies on 2D and 3D examples to demonstrate the superior performance of SGA over all existing exact Wasserstein barycenter algorithms.

On the other hand, we would like to mention some computational limitation of the current approach. The SGA algorithm is mainly designed to compute the Wasserstein barycenter of 2D and 3D distributions discretized over the grid in a compact domain. Since the grid size scales exponentially in the domain dimension, how to make the SGA approach viable to higher-dimensional domains is an open problem that we leave to future research.

However, to the best of our knowledge, all existing high-accuracy exact barycenter algorithms are limited to 2D or 3D problems. The entropic methods implemented in the POT library Flamary et al. (2021) are limited to the computation of barycenters for 2D distributions. For higher dimensional probles, existing literature either rely on nerual network based methods (Fan et al., 2025; Kolesov et al., 2024a) or target a regularized notion of barycenter (Chi et al., 2023), which trade numerical accuracy for scalability.

To extended our approach from regular to non-uniform grids, three components require modification. First, the fast $c$-transform must be replaced, for example, by adapting the parametric Legendre transform alogrithm by Hiriart-Urruty & Lucet (2007). Second, the FFT-based Poisson solver can be substituted by multigrid methods. Third, the computation of the pushforward measure, currently obtained by pushing forward discretized marginals under a discretized map, would also need multigrid techniques to handle the Jacobian. Since multigrid methods are substantially more expensive than FFT-based Poisson solvers, higher computational costs are expected.

## 1.3 NOTATIONS

Given $u, v \in \mathbb{R}^d$, we denote the standard vector inner product and the Euclidean norm by $\langle u, v \rangle$ and $\|v\|_2 = \sqrt{\langle v, v \rangle}$ respectively. Let $\Omega \subset \mathbb{R}^d$ be a compact and convex set. The set of probability measures on $\Omega$ with finite second moments is denoted by $\mathcal{P}_2(\Omega)$. For any $\mu \in \mathcal{P}_2(\Omega)$, $T_\# \mu$ represents the pushforward of $\mu$ by $T : \Omega \to \Omega$. Finally, we use $T_{\mu \to \nu}$ to denote the optimal transport map that pushes $\mu$ to $\nu$. For any $f : \Omega \to \mathbb{R}$ and continuous symmetric function $c : \Omega \times \Omega \mapsto \mathbb{R}$, the $c$-transform of $f$, denoted as $f^c : \Omega \to \mathbb{R}$, is defined as $f^c(x) := \inf_{y \in \Omega} \{ c(x, y) - f(y) \}$. In this paper, we focus on the case that $c(x, y) := \|x - y\|_2^2 / 2$. The homogeneous Sobolev space $\dot{\mathbb{H}}^1(\Omega) := \{ f : \Omega \to \mathbb{R} | \int_\Omega f \mathrm{d}x = 0 \text{ and } \|f\|_{\dot{\mathbb{H}}^1} < \infty \}$ is a Hilbert space equipped with the $\dot{\mathbb{H}}^1$-inner product $\langle f, g \rangle_{\dot{\mathbb{H}}^1} := \int_\Omega \langle \nabla f(x), \nabla g(x) \rangle \, \mathrm{d}x$ and $\dot{\mathbb{H}}^1$-norm $\|f\|_{\dot{\mathbb{H}}^1} = \sqrt{\langle f, f \rangle_{\dot{\mathbb{H}}^1}}$. To distinguish it from the standard gradient $\nabla f$, we denote the $\dot{\mathbb{H}}^1$-gradient of the functional $\mathcal{I} : \dot{\mathbb{H}}^1 \to \mathbb{R}$ as $\boldsymbol{\nabla} \mathcal{I}$, which is identified by the Riesz representation theorem via $\langle \boldsymbol{\nabla} \mathcal{I}(f), h \rangle_{\dot{\mathbb{H}}^1} = \delta \mathcal{I}_f(h)$, where $\delta \mathcal{I}_f(h)$ is the Gateaux derivative of the functional $\mathcal{I}$ at $f \in \dot{\mathbb{H}}^1$ in the direction $h$. With a slight abuse of notation, we also use $\mu$ to denote its density, and define $\|\mu\|_\infty := \max_{x \in \Omega} \mu(x)$. For any two probability densities $\mu, \nu$, the dual of $\dot{\mathbb{H}}^1$-norm is defined as $\|\mu - \nu\|_{\dot{\mathbb{H}}^{-1}} := \max \{ \int_\Omega \phi \, \mathrm{d}[\mu - \nu] : \|\phi\|_{\dot{\mathbb{H}}^1} \leqslant 1 \}$. Further information on $\dot{\mathbb{H}}^{-1}$-norm can be found in Section 5.5.2 of Santambrogio (2015).

## 2 PRELIMINARY: OPTIMAL TRANSPORT COMPUTATION

A function $f : \Omega \to \mathbb{R}$ is called $c$-concave if $f = g^c$ for some $g : \Omega \to \mathbb{R}$. In this case, $f^{cc} = f$. For any $\mu, \nu \in \mathcal{P}_2(\Omega)$ that are absolutely continuous with respect to the Lebesgue measure, the optimal transport $T_{\mu \to \nu}$ can be obtained by solving the Kantorovich dual problem,

$$\max_{f : c\text{-concave}} \left\{ \mathcal{I}(f) := \int_\Omega f \mathrm{d}\mu + \int_\Omega f^c \mathrm{d}\nu \right\}, \tag{2}$$

where the supremum is taken over all $c$-concave functions. Let $\tilde{f}$ be a maximizer, we have

$$T_{\nu \to \mu} = T_{\tilde{f}^c} \text{ and } T_{\mu \to \nu} = T_{\tilde{f}}, \text{ where } T_h := \mathrm{id} - \nabla h. \tag{3}$$

The optimal transport map $T_{\mu \to \nu}, T_{\nu \to \mu}$ are also called the Brenier maps (Brenier, 1991). Moreover, $W_2(\mu, \nu) = [2\mathcal{I}(\tilde{f})]^{1/2}$. For more details, we refer the reader to Santambrogio (2015).

Our barycenter algorithm in Section 4 is motivated by the *exact* computation of the two-marginal optimal transport problem with distributions discretized over a grid of size $n$. The $\dot{\mathbb{H}}^1$-gradient based approach proposed by Jacobs & Léger (2020) is a popular algorithm due to its numerical stability and effectiveness in handling large $n$. In particular, the $\dot{\mathbb{H}}^1$-gradient of $\mathcal{I}(f)$ is defined as

$$\boldsymbol{\nabla}\mathcal{I}(f) = g, \tag{4}$$

where $g$ is the solution to the Neumann problem (Santambrogio, 2015, Subsection 5.5.3),

$$\begin{cases} -\Delta g = \mu - (T_{f^c})_{\#}\nu, & \text{in } \Omega, \\ \frac{\partial g}{\partial \mathbf{n}} = 0, & \text{on } \partial\Omega. \end{cases} \tag{5}$$

Then, Algorithm 1, consisting of a $\dot{\mathbb{H}}^1$ gradient ascent step and a double $c$-transform step, has been introduced to solve (2) (Jacobs & Léger, 2020). Numerically, both $\boldsymbol{\nabla}\mathcal{I}(f)$ and $f^{cc}$ can be computed with time complexity $\mathcal{O}(n \log n)$ and space complexity $\mathcal{O}(n)$, assuming the probability measures have known densities on a grid of size $n$. In contrast, the exact approach via linear programming for computing optimal transport between two discrete distributions has a time complexity of $\mathcal{O}(n^3)$ and a space complexity of $\mathcal{O}(n^2)$ (Kuhn, 1955; Bertsekas & Castanon, 1989; Luenberger & Ye, 2008).

In practice, Algorithm 1 can be enhanced by a back-and-forth approach (Jacobs & Léger, 2020), where the idea is to apply the $\dot{\mathbb{H}}^1$ gradient ascent step and the double $c$-transform step iteratively by hopping between Problem (2) and its twin problem, in order to bound the Hessian in one problem by its twin. The double $c$-transform is employed for two main reasons: (1) applying the double $c$-transform to $f^{(t-\frac{1}{2})}$ does not decrease the objective value, i.e., $\mathcal{I}(f^{(t)}) = \mathcal{I}((f^{(t-\frac{1}{2})})^{cc}) \geqslant \mathcal{I}(f^{(t-\frac{1}{2})})$; (2) The double $c$-transform step ensures that $\{f^{(t)}\}_{t=1}^T$ produced by Algorithm 1 are all $c$-concave. The functions $f^c$ and $f^{cc}$ share the same regularity as $c(x, y) = \|x - y\|_2^2/2$, which is generally not the case for $f$ itself.

---

**Algorithm 1** Constrained $\dot{\mathbb{H}}^1$-Gradient Ascent Algorithm

---

Initialize $f^{(0)}$;
**for** $t = 1, 2, \ldots, T$ **do**
    $f^{(t-\frac{1}{2})} = f^{(t-1)} + \eta_t \boldsymbol{\nabla}\mathcal{I}(f^{(t-1)})$;
    $f^{(t)} = (f^{(t-\frac{1}{2})})^{cc}$;
**end for**

---

**Issues**. Despite the empirical advantages of applying the double $c$-transform, it also has theoretical drawbacks, most notably, it is not an orthogonal projection, and consequently, for any $c$-concave function $g$, the inequality $\|f^{cc} - g\|_{\dot{\mathbb{H}}^1} \leqslant \|f - g\|_{\dot{\mathbb{H}}^1}$ does not necessarily hold. As a result, Algorithm 1 is a *constrained* algorithm and the global convergence guarantee is still open, despite the fact that $\mathcal{I} : \dot{\mathbb{H}}^1 \to \mathbb{R}$ is a concave functional.

## 3 Constraint-free Sobolev gradient ascent

Before moving to the barycenter problem, we first discuss a simplified *unconstrained* $\dot{\mathbb{H}}^1$-gradient ascent algorithm for the two-marginal OT computation. This algorithm provides the backbone structure for developing our new coordinate-wise version of the barycenter optimization algorithm in Section 4.

Our starting point is the key observation that the first variation of $\mathcal{I}(f)$, and consequently $\boldsymbol{\nabla}\mathcal{I}(f)$, exist even when $f$ is not $c$-concave (cf. Proposition 2.9 in (Gangbo, 2004)), *and* the computation of $\boldsymbol{\nabla}\mathcal{I}(f^{(t)})$ at each iteration in Algorithm 1 depends solely on $T_{(f^{(t)})^c}$. Recall $f^{ccc} = f^c$ for any continuous function $f$, it follows that $\boldsymbol{\nabla}\mathcal{I}(f^{(t)})$ remains unchanged regardless of whether the double $c$-transform step is applied. Therefore, the Kantorovich dual problem (2) can be relaxed to be optimized among continuous functions while retaining the same maximum value (Santambrogio, 2015, Section 1.2); that is,

$$\arg\max\{\mathcal{I}(f) : f \text{ is } c\text{-concave}\} = \arg\max\{\mathcal{I}(f) : f \text{ is continuous}\}.$$

These together allow us to consider the unconstrained problem $\max_f \mathcal{I}(f)$ (beyond continuity) and to propose the following one-step Sobolev gradient ascent method.

---

**Algorithm 2** Sobolev Gradient Ascent (SGA) Algorithm

---

Initialize $f^{(0)}$;
**for** $t = 1, 2, \ldots, T$ **do**
    $f^{(t)} = f^{(t-1)} + \eta_{t-1} \boldsymbol{\nabla} \mathcal{I}(f^{(t-1)})$;
**end for**

---

The proposed one-step Algorithm 2 enjoys the following global convergence guarantee, with a rate that aligns with subgradient descent methods for minimizing nonsmooth convex functions in the Euclidean space; see Theorem 3.2.2 in (Nesterov, 2013) or Chapter 3.1 in (Lan, 2020).

**Theorem 1** (Convergence rate for SGA). *Let $\{f^{(t)}\}_{t=1}^T$ be the sequence computed from Algorithm 2. Assuming that $\mu$ and $\nu$ are absolutely continuous with respect to the Lebesgue measure, and $\{f^{(t)}\}_{t=1}^T$ are continuous on $\Omega$. Then,*

$$\mathcal{I}(\tilde{f}) - \mathcal{I}(f^{(\text{best})}) \leqslant \frac{\|f^{(0)} - \tilde{f}\|_{\dot{\mathbb{H}}^1}^2 + M^2 \sum_{t=1}^T \eta_{t-1}^2}{2 \sum_{t=1}^T \eta_{t-1}},$$

*where $M = \max_{t=1}^T \|\boldsymbol{\nabla}\mathcal{I}(f^{(t-1)})\|_{\dot{\mathbb{H}}^1}$, $\tilde{f}$ is any c-concave maximizer and $f^{(\text{best})}$ is the best solution found, i.e., $\mathcal{I}(f^{(\text{best})}) \geqslant \max_{t=1}^T \mathcal{I}(f^{(t)})$.*

**Remark 2.** *The boundedness of $M$ plays the role as the Lipschitz condition in Euclidean space when the objective function lacks stronger smoothness. We note that $\|\boldsymbol{\nabla}\mathcal{I}(f^{(t-1)})\|_{\dot{\mathbb{H}}^1} = \|\mu - (T_{(f^{(t-1)})^c})_{\#}\nu\|_{\dot{\mathbb{H}}^{-1}}$. The values of $r_t := \|\mu - (T_{(f^{(t-1)})^c})_{\#}\nu\|_{\dot{\mathbb{H}}^{-1}}$ computed in the implementation of Algorithm 2 for the 2D densities in Section 5.1 are reported in Table 3 (Appendix). They decrease with $t$, thus supporting the boundedness assumption on $M$ and indicating convergence to the target solution. The continuity assumption on the trajectory of updates $\{f^{(t)}\}_{t=1}^T$ is mild and may be inferred from the boundedness condition $M$, depending on dimension $d$, see Subsection A.3 in the Appendix for more details.*

## 4 WASSERSTEIN BARYCENTER COMPUTATION

Now, we are ready to introduce a new *constraint-free concave* dual formulation for the Wasserstein barycenter problem as follows:

$$\max_{f_1, \ldots, f_{m-1}} \left\{ \mathcal{D}(f_1, \ldots, f_{m-1}) := \sum_{i=1}^{m-1} \alpha_i \int_\Omega f_i^c \, \mathrm{d}\mu_i + \alpha_m \int_\Omega f_{\text{mix}}^c \, \mathrm{d}\mu_m \right\}, \tag{6}$$

where $f_{\text{mix}} = -\sum_{i=1}^{m-1} \frac{\alpha_i}{\alpha_m} f_i$ and the supremum over $f_1, \ldots, f_{m-1}$ is unconstrained beyond being continuous (cf. Lemma 8). This formulation also facilitates the analysis of the signed Wasserstein barycenter problem (Jacobs & Zhou, 2026), in which negative weights are permitted.

First, we show that the functional $\mathcal{D}$ is jointly concave.

**Lemma 3.** *The mapping $(f_1, \ldots, f_{m-1}) \mapsto \mathcal{D}(f_1, \ldots, f_{m-1})$ is jointly concave.*

The Wasserstein barycenter is unique provided with at least one marginal $\mu_i$ that is absolutely continuous (Agueh & Carlier, 2011; Kim & Pass, 2017). Our next result characterizes this unique barycenter by the maximizer to (6).

**Theorem 4** (Strong duality and barycenter characterization). *Assume that $\mu_1, \ldots, \mu_m$ are absolutely continuous with respect to the Lebesgue measure. Then the followings are true.*

(i) *If $(\tilde{f}_1, \ldots, \tilde{f}_{m-1})$ is optimal to $\mathcal{D}(f_1, \ldots, f_{m-1})$, then for any $i = 1, 2, \ldots, m-1$, $\tilde{f}_i$ is identitical to a c-concave function $\mu_i$-almost everywhere. Moreover, $\tilde{f}_{\text{mix}} := -\sum_{i=1}^{m-1} \frac{\alpha_i}{\alpha_m} \tilde{f}_i$ is identical to a c-concave function $\mu_m$-almost everywhere.*

*(ii) The strong duality holds:* $\min_{\nu \in \mathcal{P}_2(\Omega)} \mathcal{B}(\nu) = \max_{f_1,\ldots,f_{m-1}} \mathcal{D}(f_1,\ldots,f_{m-1})$.

*Let $(\tilde{f}_1,\ldots,\tilde{f}_{m-1})$ be c-concave maximizer to $\mathcal{D}(f_1,\ldots,f_{m-1})$. Then the unique Wasserstein barycenter can be obtained via the formula*

$$\tilde{\nu} = (T_{\tilde{f}_i^c})_{\#}\mu_i = (T_{\tilde{f}_{\mathrm{mix}}^c})_{\#}\mu_m. \tag{7}$$

### 4.1 SOBOLEV GRADIENT ASCENT FOR BARYCENTER COMPUTATION

We propose a Sobolev gradient ascent (SGA) approach for solving Problem (6). Our algorithm to maximize $\mathcal{D}$, as detailed in Algorithm 3, admits a global convergence guarantee and meanwhile enjoys a time complexity $O(m \times n \log(n))$ per iteration for distributions whose densities are known on the same grid of size $n$. To describe the SGA algorithm, we first derive the formula for the Sobolev gradient of $\mathcal{D}$ in the following lemma.

**Lemma 5** (Dual gradient). *Assume for $i = 1, 2, \ldots, m - 1$ that $f_i$ are continuous. Then,*

*(i) The first variation of $\mathcal{D}$ at $f_i$, denoted as $\delta\mathcal{D}_{f_i}$ can be expressed as*

$$\delta\mathcal{D}_{f_i} = -\alpha_i\big((T_{f_i^c})_{\#}\mu_i - (T_{f_{\mathrm{mix}}^c})_{\#}\mu_m\big).$$

*(ii) The $\dot{\mathbb{H}}^1$-gradient of $\mathcal{D}$ at $f_i$, denoted as $\boldsymbol{\nabla}_{f_i}\mathcal{D}$ can be expressed as*

$$\boldsymbol{\nabla}_{f_i}\mathcal{D} = (-\Delta)^{-1}\delta\mathcal{D}_{f_i} = (-\Delta)^{-1}\big(-\alpha_i\big((T_{f_i^c})_{\#}\mu_i - (T_{f_{\mathrm{mix}}^c})_{\#}\mu_m\big)\big),$$

*where $(-\Delta)^{-1}$ denotes the negative inverse Laplacian operator with zero Neumann boundary condition.*

---

**Algorithm 3** SGA Algorithm for Barycenter Computation

---

Initialize $f_i^{(0)}, i = 1, 2, \ldots, m - 1$;
**for** $t = 1, 2, \ldots, T$ **do**
    **for** $i = 1, 2, \ldots, m - 1$ **do**
        $f_i^{(t)} = f_i^{(t-1)} + \eta_{t-1}\boldsymbol{\nabla}_{f_i}\mathcal{D}(f_1^{(t-1)}, \ldots, f_{m-1}^{(t-1)})$;
    **end for**
**end for**

---

We show that the SGA Algorithm 3 enjoys the following global convergence, which is the main theoretical result of this paper.

**Theorem 6** (Main theorem: convergence rate for barycenter optimization with SGA). *Let $\{f_i^{(t)} : i = 1, \ldots, m-1\}_{t=1}^{T}$ be the sequence produced from Algorithm 3. Assuming that $\{\mu_i\}_{i=1}^{m}$ are absolutely continuous with respect to the Lebesgue measure, and that the functions $\{f_i^{(t)} : i = 1, \ldots, m-1\}_{t=1}^{T}$ are continuous on $\Omega$. Then, we have the following two scenarios.*

*(i) **Constant step size.** If the total number of iterations $T$ is fixed a priori, then by setting $M = \max_{t=1}^{T}(\sum_{i=1}^{m-1}\alpha_i^2\|(T_{(f_i^{(t-1)})^c})_{\#}\mu_i - (T_{(f_{\mathrm{mix}}^{(t-1)})^c})_{\#}\mu_m\|_{\dot{\mathbb{H}}^{-1}}^2)^{1/2}$ and $\eta_{t-1} = (\sum_{i=1}^{m-1}\|f_i^{(0)} - \tilde{f}_i\|_{\dot{\mathbb{H}}^1}^2)^{1/2}/(M\sqrt{T})$, we have*

$$\mathcal{D}(\tilde{f}_1,\ldots,\tilde{f}_{m-1}) - \mathcal{D}(f_1^{(\mathrm{best})},\ldots,f_{m-1}^{(\mathrm{best})}) \leqslant M\big(\sum_{i=1}^{m-1}\|f_i^{(0)} - \tilde{f}_i\|_{\dot{\mathbb{H}}^1}^2\big)^{1/2}\frac{1}{\sqrt{T}},$$

*where $f_{\mathrm{mix}}^{(t-1)} = -\sum_{i=1}^{m-1}\frac{\alpha_i}{\alpha_m}f_i^{(t-1)}$, $(\tilde{f}_1,\ldots,\tilde{f}_{m-1})$ is any c-concave maximizer of $\mathcal{D}$, and $\mathcal{D}(f_1^{(\mathrm{best})},\ldots,f_{m-1}^{(\mathrm{best})}) \geqslant \max_{t=1}^{T}\mathcal{D}(f_1^{(t)},\ldots,f_{m-1}^{(t)})$.*

*(ii) **Annealing step size.** If the total number of iterations $T$ is not fixed a priori, then by setting $\eta_{t-1} = (\sum_{i=1}^{m-1}\|f_i^{(0)} - \tilde{f}_i\|_{\dot{\mathbb{H}}^1}^2)^{1/2}/(M\sqrt{t})$, we have*

$$\mathcal{D}(\tilde{f}_1,\ldots,\tilde{f}_{m-1}) - \mathcal{D}(f_1^{(\mathrm{best})},\ldots,f_{m-1}^{(\mathrm{best})}) \leqslant M\big(\sum_{i=1}^{m-1}\|f_i^{(0)} - \tilde{f}_i\|_{\dot{\mathbb{H}}^1}^2\big)^{1/2}\frac{\ln(T) + 2}{\sqrt{T}}.$$

Similar to Remark 2, the continuity assumption of potential functions $\{f_i^{(t)}\}$ is mild in the literature of optimal transport theory and can be inferred from the boundedness condition that $\max_{i,t} \|(T_{(f_i^{(t-1)})^c}) \# \mu_i\| < \infty$. The two schemes on the step size choice in our Theorem 6 strongly resonates with the standard convergence results of subgradient methods for minimizing convex nonsmooth functions in $\mathbb{R}^d$. Specifically, it is known from Theorem 3.2.2 in (Nesterov, 2013) that the subgradient method with a constant step size (equivalently, with fixed iteration number $T$) is optimal for such problem uniformly in all dimension $d$ and the annealing step size scheme is sub-optimal with an extra $\ln T$ factor. Therefore, we see that Theorem 6 extends the Euclidean convergence rates to the Wasserstein barycenter optmization setting with essential structures.

**Remark 7.** *Yao et al. (2025) derived a proximal method for minimizing the general nonsmooth convex functionals in the space of probability measures based on the idea of discretizing the KL divergence gradient flow. Their coordinate KL divergence gradient descent (CKLGD) algorithm operates solely in the primal space, achieving a similar rate to our Theorem 6 and the classical subgradient methods in the Euclidean space (Nesterov, 2013; Lan, 2020). However, the computation of CKLGD relies heavily on the specialized form of the objective functional in certain entropic regularized problems by sampling. Even though the functional (1) is convex in the linear structure, it is unclear and highly nontrivial how to derive an algorithm to minimize the Wasserstein barycenter functional without entropic regularization.*

## 5   EMPIRICAL STUDY

We compare our method with the Debiased Sinkhorn Barycenter (DSB) algorithm (Janati et al., 2020), the Convolutional Wasserstein Barycenter (CWB) (Solomon et al., 2015), and the recently proposed Wasserstein-Descent $\dot{\mathbb{H}}^1$-Ascent (WDHA) algorithm (Kim et al., 2025). For DSB and CWB, we used the implementations provided in the "POT: Python Optimal Transport" library (Flamary et al., 2021), and for WDHA, we used the implementation available at `https://kaheonkim.github.io/WDHA/`. All computations are performed on a Dell PowerEdge R6525 server equipped with two 32-core AMD EPYC 7543 CPUs.

### 5.1   2D SYNTHETIC DISTRIBUTION

In this example, the goal is to compute the weighted Wasserstein baycenter of four densities supported on different shapes whose values are know on a equally spaced grid of size $1024 \times 1024$. The stepsizes of WDHA and SGA are set to be $\eta_t = 0.1$ for all $t$, and the regularization parameter of CWB and DSB is set to be reg $= 0.005$. The computed barycenters using 300 iterations are shown in Figure 2. The results of WDHA and SGA with annealing step sizes ($\eta_t = 0.5/\sqrt{t}$) are provided in Appendix, Subsection B.1. Although the barycenters computed by SGA and WDHA show no obvious visual differences between them, they exhibit clearer and sharper details compared to those from CWB and DSB. This is expected, as SGA and WDHA are exact methods based on the $\dot{\mathbb{H}}^1$-gradient, whereas CWB and DSB rely on entropic approximations and are not exact.

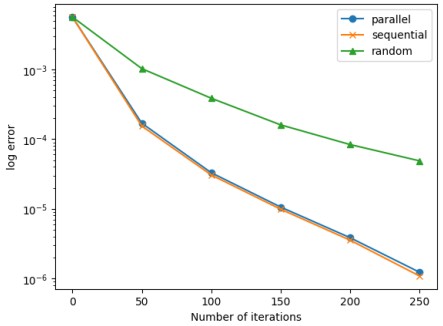

Figure 1: Empirical convergence rates of parallel (Algorithm 3), sequential (Algorithm 4), and random (Algorithm 5) SGA algorithms.

In addition, we report the computation runtime and barycenter values in Table 1, where we apply the back-and-forth approach Jacobs & Léger (2020) to compute the Wasserstein distance between the estimated barycenter and each distribution. SGA achives the smallest barycenter value for all cases. In terms of computation time, SGA requires approximately half the time of WDHA, one-third the time of CWB, and one-seventh the time of DSB. In addition, SGA is advantageous over WDHA in terms of simplicity, faster runtime, and guaranteed global convergence.

Algorithm 3 outlines a parallel scheme for computing the Wasserstein barycenter. As in (Yao et al., 2024) for coordinate-wise optimization, we also report the empirical convergence rate of this parallel scheme alongside two alternative variants: (1) SGA-S: sequential scheme outlined in Algorithm 4; (2) SGA-R: random scheme outlined in Algorithm 5. By default, we refer SGA for

Table 1: Barycenter functional value and runnting time for SGA, WDHA, CWB, and DSB for the 2D synthetic distribution example in Section 5.1. [↓] means the smaller number, the better performance. Bold numbers indicate the best performer among the four methods under comparison.

| Weights | Barycenter Functional Value ($\times 10^3$) [↓] | | | | Time (in sec) [↓] | | | |
|---|---|---|---|---|---|---|---|---|
| | SGA | WDHA | CWB | DSB | SGA | WDHA | CWB | DSB |
| $(\frac{2}{3}, 0, 0, \frac{1}{3})$ | **3.917** | 3.940 | 4.202 | 3.926 | **298** | 625 | 1346 | 3643 |
| $(\frac{1}{3}, 0, 0, \frac{2}{3})$ | **3.917** | 3.923 | 4.160 | 3.925 | **221** | 569 | 1328 | 3679 |
| $(\frac{2}{3}, \frac{1}{3}, 0, 0)$ | **5.257** | 5.275 | 5.559 | 5.260 | **257** | 602 | 1145 | 3728 |
| $(\frac{1}{3}, \frac{1}{4}, \frac{1}{6}, \frac{1}{4})$ | **5.923** | 5.952 | 6.153 | 5.936 | **700** | 1064 | 2290 | 5003 |
| $(\frac{1}{4}, \frac{1}{6}, \frac{1}{4}, \frac{1}{3})$ | **5.299** | 5.313 | 5.513 | 5.314 | **640** | 1049 | 2464 | 5192 |
| $(0, 0, \frac{1}{3}, \frac{2}{3})$ | **1.646** | 1.649 | 1.821 | 1.648 | **202** | 550 | 1262 | 3798 |
| $(\frac{1}{3}, \frac{2}{3}, 0, 0)$ | **5.257** | 5.268 | 5.513 | 5.263 | **214** | 537 | 1256 | 3885 |
| $(\frac{1}{4}, \frac{1}{3}, \frac{1}{4}, \frac{1}{6})$ | **5.964** | 5.987 | 6.182 | 5.977 | **672** | 984 | 2520 | 5062 |
| $(\frac{1}{6}, \frac{1}{4}, \frac{1}{3}, \frac{1}{4})$ | **5.219** | 5.232 | 5.424 | 5.239 | **626** | 961 | 2515 | 5076 |
| $(0, 0, \frac{2}{3}, \frac{1}{3})$ | **1.645** | 1.647 | 1.841 | 1.648 | **206** | 498 | 1255 | 3664 |
| $(0, \frac{2}{3}, \frac{1}{3}, 0)$ | **3.812** | 3.833 | 4.006 | 3.814 | **211** | 550 | 1247 | 3675 |
| $(0, \frac{1}{3}, \frac{2}{3}, 0)$ | **3.813** | 3.814 | 4.023 | 3.814 | **209** | 510 | 1251 | 3648 |

SGA with the parallel scheme. Using $(f_1^{(\text{best})}, \ldots, f_{m-1}^{(\text{best})})$ as an estimation for $(\tilde{f}_1, \ldots, \tilde{f}_{m-1})$, we plot $\log(\mathcal{D}(f_1^{(\text{best})}, \ldots, f_{m-1}^{(\text{best})}) - \mathcal{D}(f_1^{(t)}, \ldots, f_{m-1}^{(t)}))$ against $t$ in Figure 1.

---

**Algorithm 4** SGA-S

---

Initialize $f_i^{(0)}, i = 1, 2, \ldots, m - 1$;
**for** $t = 1, 2, \ldots, T$ **do**
    **for** $i = 1, 2, \ldots, m - 1$ **do**
        $f_i^{(t)} = f_i^{(t-1)} + \eta_{t-1} \boldsymbol{\nabla}_{f_i} \mathcal{D}(f_1^{(t)}, \ldots, f_{i-1}^{(t)}, f_i^{(t-1)}, \ldots, f_{m-1}^{(t-1)})$;
    **end for**
**end for**

---

**Algorithm 5** SGA-R

---

Initialize $f_i^{(0)}, i = 1, 2, \ldots, m - 1$;
**for** $t = 1, 2, \ldots, T$ **do**
    Sample $i_t$ uniformly from $\{1, 2, \ldots, m - 1\}$;
    $f_{i_t}^{(t)} = f_{i_t}^{(t-1)} + \eta_{t-1} \boldsymbol{\nabla}_{f_{i_t}} \mathcal{D}(f_1^{(t-1)}, \ldots, f_{m-1}^{(t-1)})$;
**end for**

---

## 5.2 3D INTERPOLATION

We apply Wasserstein barycenter algorithms to interpolate between a 3D ball and a 3D cube, discretized over a grid of size $200 \times 200 \times 200$. Currently, optimal transport computations for 3D distributions are not supported by the POT library (Flamary et al., 2021). In this experiment, an annealing step size of $\eta_t = 5 \times 10^{-3}/\sqrt{t}$ is used for both SGA and WDHA. The results of SGA are shown in Figure 3, with barycenter values (in $10^{-3}$) of 1.443, 1.577, and 1.455 from left to right. Interestingly, the WDHA algorithm diverges under this setting, indicating that SGA is numerically more stable. This increased stability may stem from SGA's simpler structure and its global convergence guarantees.

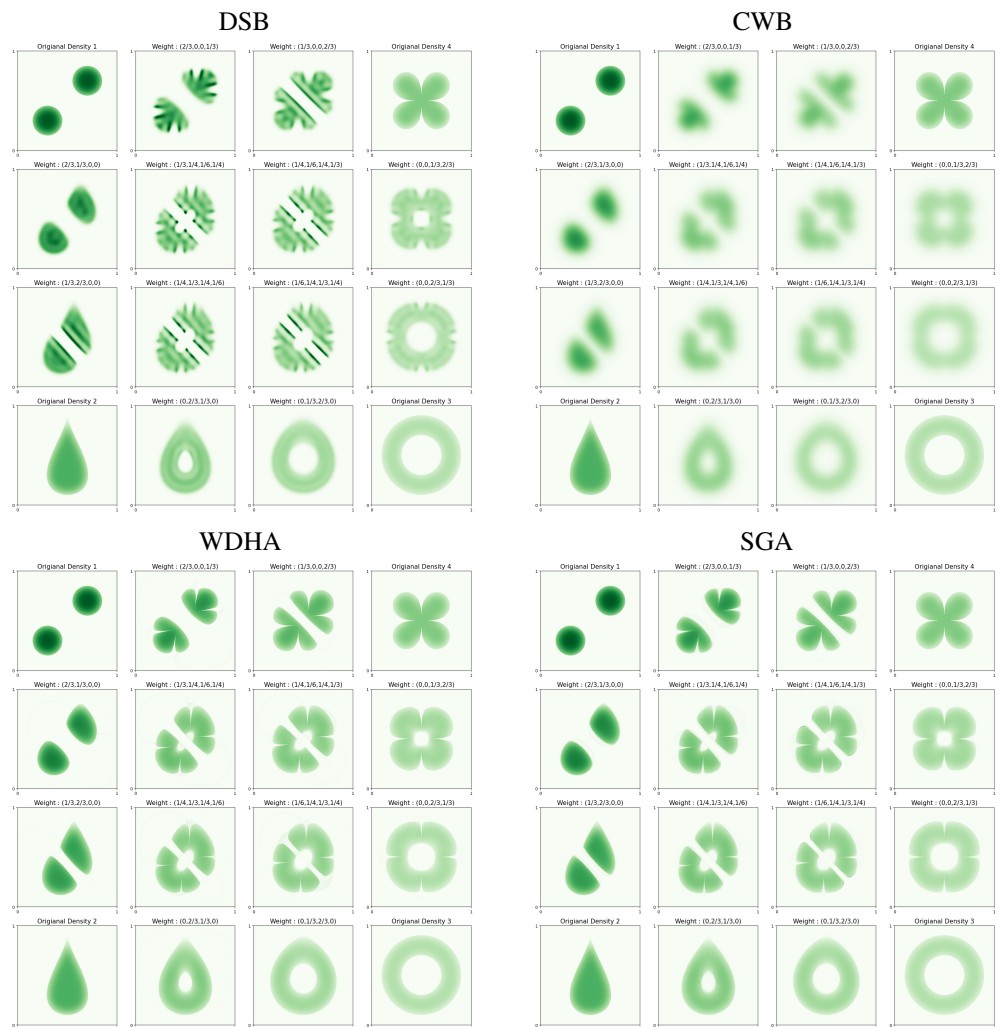

Figure 2: Comparison of weighted Wasserstein barycenter of densities supported on different shapes.

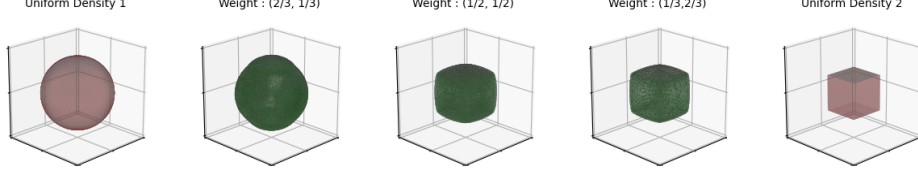

Figure 3: Interpolation using SGA between a 3D ball and a 3D cube.

## 5.3 REAL DATA EXAMPLE

The Wasserstein barycenter can be used to compress meaningful information from video frames. Such compressed representations may be valuable for tasks like object detection, motion tracking, and behavior analysis. Figure 4 (left) displays 16 frames from a video of the electric scooter tracking data, available from Kaggle (URL `https://www.kaggle.com/datasets/trainingdatapro/electric-scooters-tracking`). In these frames, two prominent moving objects are observed: a person walking toward the surveillance camera from the top right, and a car in the center making a right turn. The static elements include the grass, trees, and road. We compute the Wasserstein barycenter of the frames, treating each image as a probability density. We

set $\eta_t = 0.1$ for WDHA and SGA, and reg $= 0.0005$ for CWB and DSB. The resulting baryceters are shown in Figure 4 (right). Among the methods compared, the SGA algorithm appears to provide the most informative representation, clearly capturing the trajectories of the moving objects while preserving a sharper background for the static components with less distortion than the other three methods.

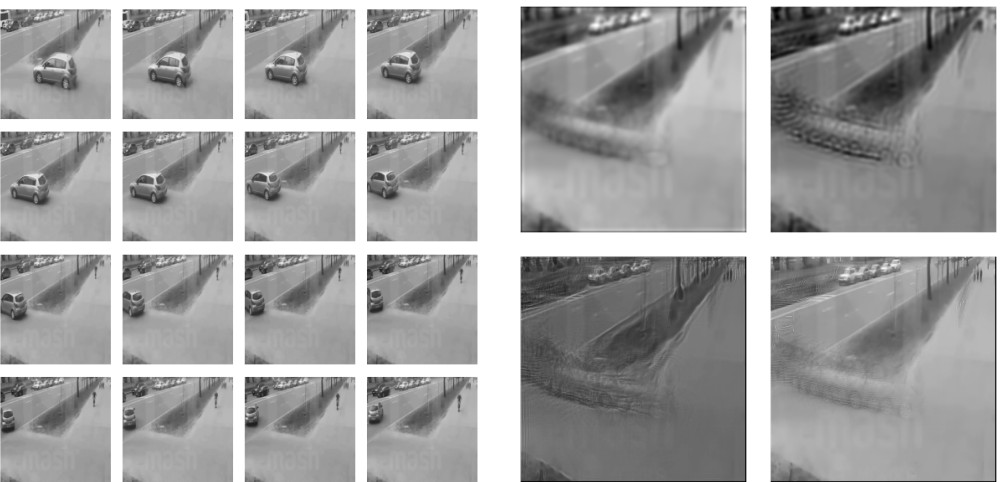

Figure 4: Left half displays 16 surveillance video frames. Right half displays their baryceters computed by CWB (top left), DSB (top right), WDHA (bottom left) and SGA (bottom right).

## REPRODUCIBILITY STATEMENT

Detailed proofs of all theorems are provided in the Appendix. The code used to produce the simulation results is available as supplementary material. The electric scooter tracking data is publicly available on Kaggle (URL https://www.kaggle.com/datasets/trainingdatapro/electric-scooters-tracking).

## ACKNOWLEDGMENTS

C. Zhu was partially supported by NSF DMS-2412832. X. Chen was partially supported by NSF DMS-2413404, NSF CAREER Award DMS-2347760, and a gift from Simons Foundation.

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

## A  TECHNICAL DETAILS

### A.1  TECHNICAL LEMMAS

In this section, all lemmas are adapted from existing literature under the specific setting where the cost function $c : \Omega \times \Omega \mapsto \mathbb{R}$ is given by $c(x, y) = \frac{1}{2}\|x - y\|_2^2$ for some convex compact subset $\Omega \in \mathbb{R}^d$. This assumption allows us to avoid introducing concepts such as the $c$-superdifferential. Specifically, Lemma 8 is adapted from Gangbo (2004); Jacobs & Léger (2020); Lemma 9 from Jacobs & Zhou (2026); Lemma 10 from Brenier (1991); Ambrosio & Gigli (2013).

**Lemma 8.** *Let $f \in C(\Omega)$ be a continuous function on a convex compact subset $\Omega \in \mathbb{R}^d$. Then it holds*

*(i) $f^{cc}(y) \geqslant f(y)$ and $f^{ccc}(x) = f^c(x)$.*

*(ii) $f^c$ is differentiable almost everywhere.*

*(iii) For any $h \in C(\Omega)$, we have $\left\|(f + \varepsilon h)^c - f^c\right\|_\infty \leqslant |\varepsilon| \|h\|_\infty$.*

*(iv) If $f^c$ is differentible at $x$, then*

$$\lim_{\varepsilon \to 0} \frac{(f + \varepsilon h)^c(x) - f^c(x)}{\varepsilon} = -h \circ T_{f^c}(x).$$

**Lemma 9.** *For any Lipschitz continuous function $\phi$, constant $a > 0$, and absolutely continuous probability measure $\mu$, we have*

$$\min_{\nu \in \mathcal{P}_2(\Omega)} \int_\Omega \phi \mathrm{d}\nu + \frac{a}{2} W_2^2(\mu, \nu) = \int_\Omega a(-\frac{\phi}{a})^c \mathrm{d}\mu.$$

**Lemma 10.** *Consider $\mu \in \mathcal{P}_2(\Omega)$ for some convex compact subset $\Omega \in \mathbb{R}^d$. If a map $T$ is in the form of $T_{f^c} = \mathrm{id} - \nabla f^c$ for some $c$-concave function $f$, then $T$ is the optimal map between $\mu$ and $(T)_{\#}\mu$. Consequently,*

$$\frac{1}{2} W_2^2(\mu, (T_{f^c})_{\#}\mu) = \int_\Omega \frac{1}{2}\left\|x - T_{f^c}(x)\right\|_2^2 \mathrm{d}\mu = \int_\Omega f^c \mathrm{d}\mu + \int_\Omega f \mathrm{d}[(T_{f^c})_{\#}\mu].$$

### A.2  PROOF TO THEOREM 1

Since $\Omega$ is compact, we can define $R := \max_{x \in \Omega} \|x\|_2^2 < \infty$. If $f$ is continuous, then by Proposition 2.9 (ii) in Gangbo (2004), for any $x_1, x_2 \in \Omega$, we have

$$|f^c(x_1) - f^c(x_2)| \leqslant 2R\|x_1 - x_2\|_2.$$

This implies that $f^c$ is $2R$-Lipschitz continuous, and hence differentiable almost everywhere on $\Omega$ with respect to the Lebesgue measure. Since $\mu$ and $\nu$ are absolutely continuous with respect to the Lebesgue measure, it follows that $f^c$ is also differentiable $\nu$-almost everywhere. By Proposition 2.9 (iii) of Gangbo (2004), the first variation of $\mathcal{I}$ at $f$, denoted $\delta \mathcal{I}_f$, is given by

$$\delta \mathcal{I}_f(h) := \int_\Omega h \, d\mu - \int_\Omega h \circ T_{f^c} \, d\nu = \langle \boldsymbol{\nabla}\mathcal{I}(f), h \rangle_{\dot{\mathbb{H}}^1}$$

for any continuous function $h$ on $\Omega$. Using the concavity of $\mathcal{I}$, we have the inequality

$$\mathcal{I}(f_2) - \mathcal{I}(f_1) - \langle \boldsymbol{\nabla}\mathcal{I}(f_1), f_2 - f_1 \rangle_{\dot{\mathbb{H}}^1} \leqslant 0.$$

Following standard subgradient descent arguments for non-smooth convex functions in Euclidean space, cf. (Nesterov, 2013), we have

$$\mathcal{I}(\tilde{f}) - \mathcal{I}(f^{(\mathrm{best})}) \leqslant \frac{\|f^{(0)} - \tilde{f}\|_{\dot{\mathbb{H}}^1}^2 + M^2 \sum_{t=1}^T \eta_{t-1}^2}{2 \sum_{t=1}^T \eta_{t-1}},$$

where $M = \max_{t=1}^T \|\mu - (T_{(f^{(t-1)})^c})_{\#}\nu\|_{\dot{\mathbb{H}}^{-1}}$.

### A.3 SUPPLEMENT TO REMARK 2

The following lemma summarize results (Mikha˘ ilov, 1978, Theorem 4 on page 217) and (Evans, 2010, page 366) on the regularity of a solution to the Poisson equation with zero Neumann boundary condition.

**Lemma 11.** *Consider a bounded open subset $\Omega \subset \mathbb{R}^d$, assume $\rho \in \mathbb{H}^k(\Omega)$ with $\int_\Omega \rho \, dx = 0$ and $\partial\Omega \in C^{k+2}$ for certain $k \geqslant 0$, then the solution $g$ to the Poisson equation with zero Neumann boundary condition belongs to $\mathbb{H}^{k+2}(\Omega)$:*

$$\begin{cases} \Delta g = \rho & \text{in } \Omega; \\ \dfrac{\partial g}{\partial n} = 0 & \text{on } \partial\Omega. \end{cases}$$

**Lemma 12** (Morrey's inequality). *Consider a bounded open subset $\Omega \subset \mathbb{R}^d$, assume $g \in \mathbb{H}^{k+2}(\Omega)$. If $(k+2) > \frac{d}{2}$, then $g \in C^{k+2-\lceil \frac{d}{2} \rceil, \gamma}$, where $\gamma = \begin{cases} \lceil \dfrac{d}{2} \rceil - \dfrac{d}{2}, & \dfrac{d}{2} \notin \mathbb{Z}; \\ \text{any element in } (0,1), & \dfrac{d}{2} \in \mathbb{Z}. \end{cases}$ Here $\lceil \frac{d}{2} \rceil$ denotes ceiling integer of $\frac{d}{2}$.*

In Theorem 1, we assume that the trajectory $\{f^{(t)}\}_{t=1}^T$ are continuous functions. Recalling the iterative update in Algorithm 2, we just need a simple induction to show that $\nabla\mathcal{I}(f^{(t-1)})$ is continuous whenever $f^{(t-1)}$ is continuous. Note that $\nabla\mathcal{I}(f^{(t-1)})$ is a solution to the Neumann problem (see equation 4 and equation 5), thus we may combine the above lemmas to propose alternative assumption ensuring continuity of the Poisson solution.

Specifically, let $\Omega \subset \mathbb{R}^d$ be a bounded open subset. To ensure $g$ used in above lemmas continuous, it suffices to require $\rho \in \mathbb{H}^k(\Omega)$ with $\int_\Omega \rho \, dx = 0$ and $\partial\Omega \in C^{k+2}$ for any integer $k > \frac{d}{2} - 2$.

For example,

- In 1D, $k = -1$ is sufficient. The continuity assumption can be replaced by $\partial\Omega \in C^1$ and $\left\| \nabla\mathcal{I}(f^{(t)}) \right\|_{\dot{\mathbb{H}}^1} = \left\| \mu - (T_{(f^{(t)})^c})_\# \nu \right\|_{\dot{\mathbb{H}}^{-1}}$ remains bounded.

- In 2D or 3D, $k = 0$ is sufficient. The continuity assumption can be replaced by $\partial\Omega \in C^2$ and $\left\| \mu - (T_{(f^{(t)})^c})_\# \nu \right\|_2$ remains bounded. In particular, given initial distributions $\mu, \nu$ with $L_2$ densities, one just needs to check the density $(T_{(f^{(t)})^c})_\# \nu$ to stay $L_2$ bounded.

### A.4 PROOF TO LEMMA 3

We first show that $f \mapsto f^c$ is concave. For any $\hat{f}$, $\bar{f}$, and $t \in [0,1]$,

$$[(1-t)\hat{f} + t\bar{f}]^c(x) = \inf_{y \in \Omega} \frac{1}{2}\|x - y\|_2^2 - (1-t)\hat{f}(y) - t\bar{f}(y)$$

$$= \inf_{y \in \Omega} (1-t)\left[\frac{1}{2}\|x-y\|_2^2 - \hat{f}(y)\right] + t\left[\frac{1}{2}\|x-y\|_2^2 - \bar{f}(y)\right]$$

$$\geqslant (1-t)(\hat{f})^c(x) + t(\bar{f})^c(x).$$

To show $\mathcal{D}(f_1, \ldots, f_{m-1})$ is jointly concave, we just need to show $(f_1, \ldots, f_{m-1}) \mapsto (-\sum_{i=1}^{m-1} \frac{\alpha_i}{\alpha_m} f_i)^c$ is concave, as the other terms are separable.

$$
\left[ -\sum_{i=1}^{m-1} \frac{\alpha_i}{\alpha_m} ((1-t)\hat{f}_i + t\bar{f}_i) \right]^c (x)
$$

$$
= \inf_{y \in \Omega} \frac{1}{2} \|x - y\|_2^2 + \sum_{i=1}^{m-1} \frac{\alpha_i}{\alpha_m} ((1-t)\hat{f}_i(y) + t\bar{f}_i(y))
$$

$$
\geqslant (1-t) \left[ \inf_{y \in \Omega} \frac{1}{2} \|x - y\|_2^2 + \sum_{i=1}^{m-1} \frac{\alpha_i}{\alpha_m} \hat{f}_i(y) \right] + t \left[ \inf_{y \in \Omega} \frac{1}{2} \|x - y\|_2^2 + \sum_{i=1}^{m-1} \frac{\alpha_i}{\alpha_m} \bar{f}_i(y) \right]
$$

$$
= (1-t) \left( -\sum_{i=1}^{m-1} \frac{\alpha_i}{\alpha_m} \hat{f}_i \right)^c (x) + t \left( -\sum_{i=1}^{m-1} \frac{\alpha_i}{\alpha_m} \bar{f}_i \right)^c (x).
$$

### A.5 PROOF TO THEOREM 4

(i) We first observe that $c$-transform is ordering reverse, i.e, suppose $f(x) \geqslant g(x)$ for any $x \in \Omega$, then

$$
f^c(x) = \inf_{y \in \Omega} \frac{1}{2} \|x - y\|_2^2 - f(y) \leqslant \inf_{y \in \Omega} \frac{1}{2} \|x - y\|_2^2 - g(y) \leqslant g^c(x).
$$

Recall for any $f$, one has $f^{cc} \geqslant f$ and $f^{ccc} = f^c$. Suppose $\tilde{f}_1^{cc} > \tilde{f}_1$ on a set of positive measure, then $f_{\text{mix}} := -\frac{\alpha_1}{\alpha_m} \tilde{f}_1^{cc} - \sum_{i=2}^{m-1} \frac{\alpha_i}{\alpha_m} \tilde{f}_i < \tilde{f}_{\text{mix}} = -\sum_{i=1}^{m-1} \frac{\alpha_i}{\alpha_m} \tilde{f}_i$ on a set of positive measure. then note that

$$
\int_\Omega \tilde{f}_1^c \, \mathrm{d}\mu_1 = \int_\Omega \tilde{f}_1^{ccc} \, \mathrm{d}\mu_1;
$$

$$
\int_\Omega f_{\text{mix}}^c \, \mathrm{d}\mu_m \geqslant \int_\Omega \tilde{f}_{\text{mix}}^c \, \mathrm{d}\mu_m.
$$

Consequently, $\mathcal{D}(\tilde{f}_1^{cc}, \tilde{f}_2, \ldots, \tilde{f}_{m-1}) \geqslant \mathcal{D}(\tilde{f}_1, \ldots, \tilde{f}_{m-1})$. Repeating these steps, we may replace the maximizer $(\tilde{f}_1, \ldots, \tilde{f}_{m-1})$ by $(\tilde{f}_1^{cc}, \ldots, \tilde{f}_{m-1}^{cc})$, which is $c$-concave. As a result, one can always select a $c$-concave maximizer, still denoted by $(\tilde{f}_1, \ldots, \tilde{f}_{m-1})$.

Assume that $\tilde{f}_{\text{mix}} = -\sum_{i=1}^{m-1} \frac{\alpha_i}{\alpha_m} \tilde{f}_i$ is not $c$-concave, that is, $\tilde{f}_{\text{mix}}^{cc} > \tilde{f}_{\text{mix}}$ on a set of positive measure. Now we replace $\hat{f}_1$ by $f_1^{(1)} := \tilde{f}_1 - \frac{\alpha_m}{\alpha_1} (\tilde{f}_{\text{mix}}^{cc} - \tilde{f}_{\text{mix}}) < \tilde{f}_1$, resulting in

$$
(f_1^{(1)})^c \geqslant \tilde{f}_1^c;
$$

$$
f_{\text{mix}}^{(1)} := -\frac{\alpha_1}{\alpha_m} f_1^{(1)} - \sum_{i=2}^{m-1} \frac{\alpha_i}{\alpha_m} \tilde{f}_i = -\frac{\alpha_1}{\alpha_m} f_1^{(1)} + \frac{\alpha_1}{\alpha_m} \tilde{f}_1 - \sum_{i=1}^{m-1} \frac{\alpha_i}{\alpha_m} \tilde{f}_i
$$

$$
= -\frac{\alpha_1}{\alpha_m} [-\frac{\alpha_m}{\alpha_1} (\tilde{f}_{\text{mix}}^{cc} - \tilde{f}_{\text{mix}})] + \tilde{f}_{\text{mix}} = \tilde{f}_{\text{mix}}^{cc}.
$$

Consequently, $\mathcal{D}(f_1^{(1)}, \tilde{f}_2, \ldots, \tilde{f}_{m-1}) \geqslant \mathcal{D}(\tilde{f}_1, \tilde{f}_2, \ldots, \tilde{f}_{m-1})$. Since $(\tilde{f}_1, \ldots, \tilde{f}_{m-1})$ is a maximizer to $\mathcal{D}(f_1, \ldots, f_{m-1})$, which yields that $f_1^{(1)}$ is identical to a $c$-concave function $\tilde{f}_1$ $\mu_1$-almost everywhere, while $f_{\text{mix}}^{(1)}$ is $c$-concave. As a result, we may select a $c$-concave maximizer, still denoted by $(\tilde{f}_1, \cdots, \tilde{f}_{m-1})$, such that $\tilde{f}_{\text{mix}} = -\sum_{i=1}^{m-1} \frac{\alpha_i}{\alpha_m} \tilde{f}_i$ is identical to a $c$-concave function $\mu_m$-almost everywhere.

(ii) We show $\min_{\nu \in \mathcal{P}_2(\Omega)} \mathcal{B}(\nu) \geqslant \max_{f_1,\ldots,f_{m-1}} \mathcal{D}(f_1,\ldots,f_{m-1})$ by noting that

$$
\begin{aligned}
\min_{\nu \in \mathcal{P}_2(\Omega)} \mathcal{B}(\nu) &= \min_{\nu \in \mathcal{P}_2(\Omega)} \sum_{i=1}^{m} \alpha_i W_2^2(\mu_i, \nu) \\
&= \min_{\nu \in \mathcal{P}_2(\Omega)} \sum_{i=1}^{m-1} \alpha_i [\max_{f_i} \int_\Omega f_i^c \, \mathrm{d}\mu_i + \int_\Omega f_i \, \mathrm{d}\nu] + \alpha_m W_2^2(\mu_m, \nu) \\
&\geqslant \max_{f_1,\ldots,f_{m-1}} \sum_{i=1}^{m-1} \int_\Omega \alpha_i f_i^c \, \mathrm{d}\mu_i + \min_{\nu \in \mathcal{P}_2(\Omega)} \int_\Omega [\sum_{i=1}^{m-1} \alpha_i f_i] \, \mathrm{d}\nu + \alpha_m W_2^2(\mu_m, \nu) \\
&= \max_{f_1,\ldots,f_{m-1}} \sum_{i=1}^{m-1} \int_\Omega \alpha_i f_i^c \, \mathrm{d}\mu_i + \alpha_m \int_\Omega (-\sum_{i=1}^{m-1} \frac{\alpha_i}{\alpha_1} f_i)^c \, \mathrm{d}\mu_m,
\end{aligned}
$$

where the last equality is due to Lemma 9.

To complete the strong duality, let us choose a $c$-concave maximizer $(\tilde{f}_1, \ldots, \tilde{f}_{m-1})$ such that $\tilde{f}_{\mathrm{mix}} = -\sum_{i=1}^{m-1} \frac{\alpha_i}{\alpha_m} \tilde{f}_i$ is $c$-concave as well. We will show that $\mathcal{D}(\tilde{f}_1, \ldots, \tilde{f}_{m-1}) = \mathcal{B}(\tilde{\nu})$, where $\tilde{\nu} = (T_{\tilde{f}_i^c})_{\#} \mu_i = (T_{\tilde{f}_{\mathrm{mix}}^c})_{\#} \mu_m$, is defined in (7). Thanks to Lemma 10, $T_{\tilde{f}_i^c}$ is the optimal map between $\mu_i$ and $\tilde{\nu} = (T_{\tilde{f}_i^c})_{\#} \mu_i$ and $T_{\tilde{f}_{\mathrm{mix}}^c}$ is the optimal map between $\mu_m$ and $\tilde{\nu} = (T_{\tilde{f}_{\mathrm{mix}}^c})_{\#} \mu_m$. That is,

$$
\begin{aligned}
\frac{\alpha_i}{2} W_2^2(\mu_i, \tilde{\nu}) &= \alpha_i [\int_\Omega \tilde{f}_i^c \, \mathrm{d}\mu_i + \int_\Omega \tilde{f}_i \, \mathrm{d}\tilde{\nu}], \qquad \text{for } i = 1, \ldots, m-1; \\
\frac{\alpha_m}{2} W_2^2(\mu_m, \tilde{\nu}) &= \alpha_m [\int_\Omega \tilde{f}_{\mathrm{mix}}^c \, \mathrm{d}\mu_m + \int_\Omega \tilde{f}_{\mathrm{mix}} \, \mathrm{d}\tilde{\nu}].
\end{aligned}
$$

Consequently,

$$
\begin{aligned}
\mathcal{D}(\tilde{f}_1, \ldots, \tilde{f}_{m-1}) &= \sum_{i=1}^{m-1} \alpha_i \int_\Omega \tilde{f}_i^c \mathrm{d}\mu_i + \alpha_m \int_\Omega \tilde{f}_{\mathrm{mix}}^c \, \mathrm{d}\mu_m \\
&= \sum_{i=1}^{m-1} \alpha_i \int_\Omega \tilde{f}_i^c \mathrm{d}\mu_i + \alpha_m \int_\Omega \tilde{f}_{\mathrm{mix}}^c \, \mathrm{d}\mu_m - \sum_{i=1}^{m} \frac{\alpha_i}{2} W_2^2(\mu_i, \tilde{\nu}) + \mathcal{B}(\tilde{\nu}) \\
&= -\sum_{i=1}^{m-1} \alpha_i \tilde{f}_i \, \mathrm{d}\tilde{\nu} - \alpha_m \int_\Omega \tilde{f}_{\mathrm{mix}} \, \mathrm{d}\tilde{\nu} + \mathcal{B}(\tilde{\nu}) \\
&= \mathcal{B}(\tilde{\nu}),
\end{aligned}
$$

where the last equality is due to the fact that $-\sum_{i=1}^{m-1} \alpha_i \tilde{f}_i - \alpha_m \tilde{f}_{\mathrm{mix}} = -\sum_{i=1}^{m-1} \alpha_i \tilde{f}_i - \alpha_m(-\sum_{i=1}^{m-1} \frac{\alpha_i}{\alpha_m} \tilde{f}_i) = 0$.

Now, $\max_{f_1,\ldots,f_{m-1}} \mathcal{D}(f_1,\ldots,f_{m-1}) \geqslant \mathcal{D}(\tilde{f}_1,\ldots,\tilde{f}_{m-1}) = \mathcal{B}(\tilde{\nu}) \geqslant \min_{\nu \in \mathcal{P}_2(\Omega)} \mathcal{B}(\nu)$, which completes the strong duality.

### A.6 PROOF TO LEMMA 5

(i) We first calculate the first variation of $(-\frac{\alpha_i}{\alpha_m} f_i - \sum_{j \neq i} \frac{\alpha_j}{\alpha_m} f_j)^c$.

Let $u = -\sum_{j \neq i} \frac{\alpha_j}{\alpha_m} f_j$, apply Lemma 8:

$$
\lim_{\varepsilon \to 0} \frac{(-\frac{\alpha_i}{\alpha_m} f_i + u - \frac{\alpha_i}{\alpha_m} \varepsilon h)^c - (-\frac{\alpha_i}{\alpha_m} f_i + u)^c}{\varepsilon} = -(-\frac{\alpha_i}{\alpha_m} h) \circ T_{(-\frac{\alpha_i}{\alpha_m} f_i + u)^c} = \frac{\alpha_i}{\alpha_m} h \circ T_{f_{\mathrm{mix}}^c}
$$

Apply this result, the Gateaux derivative $\delta\mathcal{D}_{f_i}(h)$ with fixed other inputs $f_{j\neq i}$ is given by

$$\delta\mathcal{D}_{f_i}(h)$$

$$=\lim_{\varepsilon\to 0}\alpha_i\int_\Omega \frac{(f_i+\varepsilon h)^c - f^c}{\varepsilon}\,\mathrm{d}\mu_i + \alpha_m\int_\Omega \frac{(-\frac{\alpha_i}{\alpha_m}f_i + u - \frac{\alpha_i}{\alpha_m}\varepsilon h)^c - (-\frac{\alpha_i}{\alpha_m}f_i + u)^c}{\varepsilon}\,\mathrm{d}\mu_m$$

$$=\alpha_i\int (-h\circ T_{f_i^c})\,\mathrm{d}\mu_i + \alpha_m\int \frac{\alpha_i}{\alpha_m} h\circ T_{f_{\mathrm{mix}}^c}\,\mathrm{d}\mu_m$$

$$=-\alpha_i\int h\,\mathrm{d}[(T_{f_i^c})_\#\mu_i - (T_{f_{\mathrm{mix}}^c})_\#\mu_m].$$

Consequently, the first variation $\delta\mathcal{D}_{f_i} = -\alpha_i((T_{f_i^c})_\#\mu_i - (T_{f_{\mathrm{mix}}^c})_\#\mu_m)$.

(ii) is a consequence from the fact that the $\dot{\mathbb{H}}^1$-gradient is defined via $\langle \nabla_{f_i}\mathcal{D}(f_i; f_{j\neq i}), h\rangle_{\dot{\mathbb{H}}^1} = \delta\mathcal{D}_{f_i}(h)$. Similarly with equation 4 and equation 5, we obtain that

$$\boldsymbol{\nabla}_{f_i}\mathcal{D} = (-\Delta)^{-1}\big(-\alpha_i\big((T_{f_i^c})_\#\mu_i - (T_{f_{\mathrm{mix}}^c})_\#\mu_m\big)\big).$$

## A.7 Proof to Theorem 6

We note that $\mathcal{D}$ is a functional over the product space $(\dot{\mathbb{H}}^1)^{m-1} := \dot{\mathbb{H}}^1 \times \cdots \times \dot{\mathbb{H}}^1$. For any $\mathbf{f} = (f_1,\ldots,f_{m-1})$ and $\mathbf{g} = (g_1,\ldots,g_{m-1})$, $(\dot{\mathbb{H}}^1)^{m-1}$ is a Hilbert space with inner product

$$\langle \mathbf{f}, \mathbf{g}\rangle_{(\dot{\mathbb{H}}^1)^{m-1}} := \sum_{i=1}^{m-1}\langle f_i, g_i\rangle_{\dot{\mathbb{H}}^1}.$$

We let $\boldsymbol{\nabla}\mathcal{D}(\mathbf{f}) = (\boldsymbol{\nabla}\mathcal{D}_{f_1}(\mathbf{f}),\ldots,\boldsymbol{\nabla}\mathcal{D}_{f_{m-1}}(\mathbf{f}))$. Using the concavity of $\mathcal{D}$ and the concave inequality $\mathcal{D}(\mathbf{g}) - \mathcal{D}(\mathbf{f}) - \langle \boldsymbol{\nabla}\mathcal{D}(\mathbf{f}), \mathbf{g}-\mathbf{f}\rangle_{(\dot{\mathbb{H}}^1)^{m-1}} \leqslant 0$, it follows similarly from Theorem 1 and standard analysis of subgradient descent that

$$\mathcal{D}(\tilde{f}_1,\ldots,\tilde{f}_{m-1}) - \mathcal{D}(f_1^{(\text{best})},\ldots,f_{m-1}^{(\text{best})}) \leqslant \frac{\sum_{i=1}^{m-1}\|f_i^{(0)} - \tilde{f}_i\|_{\dot{\mathbb{H}}^1}^2 + M^2\sum_{t=1}^T \eta_{t-1}^2}{2\sum_{t=1}^T \eta_{t-1}},$$

where $M = \max_{t=1}^T(\sum_{i=1}^{m-1}\alpha_i^2\|(T_{(f_i^{(t-1)})^c})_\#\mu_i - (T_{(f_{\mathrm{mix}}^{(t-1)})^c})_\#\mu_m\|_{\dot{\mathbb{H}}^{-1}}^2)^{1/2}$. If we fix the total number of iterations $T$ a priori, the optimal step size that minimize the right hand side is $\eta_{t-1} = (\sum_{i=1}^{m-1}\|f_i^{(0)} - \tilde{f}_i\|_{\dot{\mathbb{H}}^1}^2)^{1/2}/(M\sqrt{T})$. This gives us

$$\mathcal{D}(\tilde{f}_1,\ldots,\tilde{f}_{m-1}) - \mathcal{D}(f_1^{(\text{best})},\ldots,f_{m-1}^{(\text{best})}) \leqslant \frac{M(\sum_{i=1}^{m-1}\|f_i^{(0)} - \tilde{f}_i\|_{\dot{\mathbb{H}}^1}^2)^{1/2}}{\sqrt{T}}.$$

If we don't fix $T$ a priori, then by letting $\eta_{t-1} = (\sum_{i=1}^{m-1}\|f_i^{(0)} - \tilde{f}_i\|_{\dot{\mathbb{H}}^1}^2)^{1/2}/(M\sqrt{t})$, we have

$$\mathcal{D}(\tilde{f}_1,\ldots,\tilde{f}_{m-1}) - \mathcal{D}(f_1^{(\text{best})},\ldots,f_{m-1}^{(\text{best})}) \leqslant M(\sum_{i=1}^{m-1}\|f_i^{(0)} - \tilde{f}_i\|_{\dot{\mathbb{H}}^1}^2)^{1/2}\frac{\log(T)+2}{\sqrt{T}},$$

where we used inequlities $\sum_{t=1}^T \frac{1}{t} \leqslant \ln(T)+1$ and $\sum_{t=1}^T \frac{1}{\sqrt{t}} \geqslant \sqrt{T}$.

## A.8 Proof to Lemma 9

Given that $\mu$ is absolutely continuous, for any $\nu \in \mathcal{P}_2(\Omega)$, there exists a measurable map $T$ that pushes $\mu$ to $\nu$ (Santambrogio, 2015, Corollary 1.29). Thus,

$$\min_{\nu\in\mathcal{P}_2(\Omega)}\int \phi\,\mathrm{d}\nu + \frac{a}{2}W_2^2(\mu,\nu) = a\min_{\nu\in\mathcal{P}_2(\Omega)}\int \frac{\phi}{a}\,\mathrm{d}\nu + \frac{1}{2}W_2^2(\mu,\nu)$$

$$= a\inf_T \int \frac{\phi}{a}\,\mathrm{d}[(T)_\#\mu] + \int \frac{1}{2}\|x - T(x)\|_2^2\,\mathrm{d}\mu$$

$$= a\inf_T \int \big[\frac{\phi(T(x))}{a} + \frac{1}{2}\|x - T(x)\|_2^2\big]\,\mathrm{d}\mu$$

$$= a\int \inf_{y=T(x)}\big[\frac{1}{2}\|x - y\|_2^2 + \frac{\phi(y)}{a}\big]\,\mathrm{d}\mu = a\int (-\frac{\phi}{a})^c(x)\,\mathrm{d}\mu.$$

Table 2: Barycenter functional value and runnting time for SGA and WDHA using annealing step sizes for the example in Section 5.1.

| Weights | Barycenter Functional Value ($\times 10^3$) [↓] | | | | Time (in sec) [↓] | | | |
|---|---|---|---|---|---|---|---|---|
| | SGA | WDHA | CWB | DSB | SGA | WDHA | CWB | DSB |
| $(\frac{2}{3}, 0, 0, \frac{1}{3})$ | **3.917** | 4.012 | 4.202 | 3.926 | **328** | 497 | 1346 | 3643 |
| $(\frac{1}{3}, 0, 0, \frac{2}{3})$ | **3.915** | 3.931 | 4.160 | 3.925 | **215** | 458 | 1328 | 3679 |
| $(\frac{1}{3}, \frac{1}{4}, \frac{1}{6}, \frac{1}{4})$ | **5.917** | 5.999 | 6.153 | 5.936 | **678** | 822 | 2290 | 5003 |
| $(\frac{2}{3}, \frac{1}{3}, 0, 0)$ | **5.256** | 5.340 | 5.559 | 5.260 | **280** | 477 | 1145 | 3728 |
| $(\frac{1}{4}, \frac{1}{6}, \frac{1}{4}, \frac{1}{3})$ | **5.288** | 5.323 | 5.513 | 5.314 | **639** | 811 | 2464 | 5192 |
| $(0, 0, \frac{1}{3}, \frac{2}{3})$ | **1.646** | 1.646 | 1.821 | 1.648 | **201** | 447 | 1262 | 3798 |
| $(\frac{1}{3}, \frac{2}{3}, 0, 0)$ | **5.256** | 5.263 | 5.513 | 5.263 | **224** | 470 | 1256 | 3885 |
| $(\frac{1}{4}, \frac{1}{3}, \frac{1}{4}, \frac{1}{6})$ | **5.958** | 6.019 | 6.182 | 5.977 | **690** | 822 | 2520 | 5062 |
| $(\frac{1}{6}, \frac{1}{4}, \frac{1}{3}, \frac{1}{4})$ | **5.205** | 5.231 | 5.424 | 5.239 | **646** | 814 | 2515 | 5076 |
| $(0, 0, \frac{2}{3}, \frac{1}{3})$ | **1.646** | 1.648 | 1.841 | 1.648 | **218** | 440 | 1255 | 3664 |
| $(0, \frac{2}{3}, \frac{1}{3}, 0)$ | **3.812** | 3.871 | 4.006 | 3.814 | **211** | 452 | 1247 | 3675 |
| $(0, \frac{1}{3}, \frac{2}{3}, 0)$ | **3.813** | 3.817 | 4.023 | 3.814 | **219** | 433 | 1251 | 3648 |

# B  ADDITIONAL NUMERICAL RESULTS

## B.1  2D SYNTHETIC EXAMPLE WITH ANNEALING STEP SIZE

We examine the annealing step sizes ($\eta_t = 0.5/\sqrt{t}$) for SGA and WDHA using the synthetic example introduced in Subsection 5.1. The computed barycenters are shown in Figure 5, with corresponding barycenter values and computation times reported in Table 2. SGA continues to outperform the other methods. In certain cases, such as when $\text{weights} = (1/4, 1/6, 1/4, 1/3)$, visual inspection suggests that the algorithm has not yet converged. This observation aligns with our expectations based on Theorem 6, which indicates that for fixed total number of iterations ($T = 300$), constant step sizes are optimal.

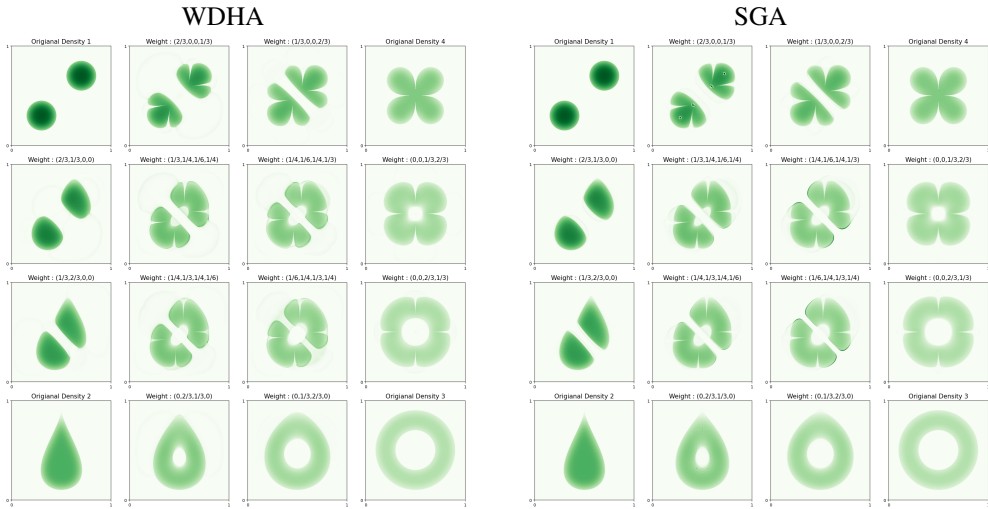

Figure 5: Comparison of weighted Wasserstein barycenter of densities supported on different shapes.

Table 3: $r_t \times 10^3$ computed in the implementation of Algorithm 2 with 2D densities in Section 5.1. The first column refers to marginals used to compute OT.

| $(\mu, \nu)$ | $t = 100$ | $t = 200$ | $t = 300$ |
|---|---|---|---|
| $(1, 2)$ | 2.459 | 1.943 | 1.813 |
| $(1, 3)$ | 1.411 | 0.952 | 0.817 |
| $(1, 4)$ | 2.348 | 1.915 | 1.820 |
| $(2, 1)$ | 2.320 | 1.357 | 1 |
| $(2, 3)$ | 0.666 | 0.397 | 0.300 |
| $(2, 4)$ | 1.169 | 0.737 | 0.573 |
| $(3, 1)$ | 2.909 | 1.792 | 1.368 |
| $(3, 2)$ | 2.061 | 1.236 | 0.924 |
| $(3, 4)$ | 1.008 | 0.593 | 0.436 |
| $(4, 1)$ | 2.930 | 1.838 | 1.435 |
| $(4, 2)$ | 1.715 | 1.155 | 0.958 |
| $(4, 3)$ | 0.593 | 0.388 | 0.331 |

## B.2 BOUNDEDNESS OF $M$ IN THEOREM 1

The radius of the bounded ball (i.e., $M$ in Theorem 1) can be easily computed and the boundedness checked in the process. Further more, $r_t := \|\mu - (T_{(f^{(t-1)})^c})_\# \nu\|_{\dot{\mathbb{H}}^{-1}}$ is expected to decrease as $t$ increases, as it quantifies the distance between $\mu$ and the pushforward measure $\nu$ by $T_{(f^{(t-1)})^c}$. If $f^{(t-1)}$ is optimal for $\mathcal{I}(f) := \int_\Omega f \, d\mu + \int_\Omega f^c \, d\nu$, we have $m_t = 0$. Empirically, we compute $m_t$ at different $t$ for pairwise 2D densities in Section 5.1, shown in Table 3. The results confirm that $r_t$ decreases with $t$, indicating that the boundedness assumption of $M$ is true and the solution will converge to the truth.

---

**Algorithm 6** Adagrad-adapted SGA for barycenter computation

---

**Require:** Stepsize $\eta > 0$, small constant $\varepsilon > 0$

  Initialize $f_i^{(0)}$ and $r_i^{(0)} \leftarrow 0$ for $i = 1, \dots, m - 1$.
  **for** $t = 1, \dots, T$ **do**
    **for** $i = 1, \dots, m - 1$ **do**
      $g_i^{(t)} \leftarrow \nabla_{f_i} \mathcal{D}\big(f_1^{(t-1)}, \dots, f_{m-1}^{(t-1)}\big)$
      $r_i^{(t)} \leftarrow r_i^{(t-1)} + g_i^{(t)} \odot g_i^{(t)}$
      $f_i^{(t)} \leftarrow f_i^{(t-1)} + \eta \dfrac{g_i^{(t)}}{\sqrt{r_i^{(t)}} + \varepsilon}$
    **end for**
  **end for**

---

## B.3 HANDWRITTEN DIGITS DATA

The handwritten digits data is commonly employed to access the performance of barycenter algorithms, see Cuturi & Doucet (2014); Ge et al. (2019); Kim et al. (2025) among others. We compare the proposed methods on high-resolution handwritten digit data from Beaulac & Rosenthal (2022), treating each image as a probability density supported on $[0, 1]^2$. Specifically, we compute the Wasserstein barycenter of 10 images of the digit 2, each of size $500 \times 500$ pixels. For SGA and WDHA, we use a constant step size $\eta_t = 0.1$. We also adapt the SGA to the AdaGrad scheme in our barycenter computation problem, yielding the SGA-AdaGrad described in Algorithm 6. For SGA-Adagrad, we set $\eta = 0.1$ and $\epsilon = 10^{-8}$. The regularization parameters for CWB and DSB are both chosen as $5 \times 10^{-3}$. The results are shown in Table 4 and Figure 6. Among all methods, SGA,

and similarly for SGA-Adagrad, produce the barycenter with the finest visible texture, while also achieving the lowest barycenter functional value and the shortest computation time.

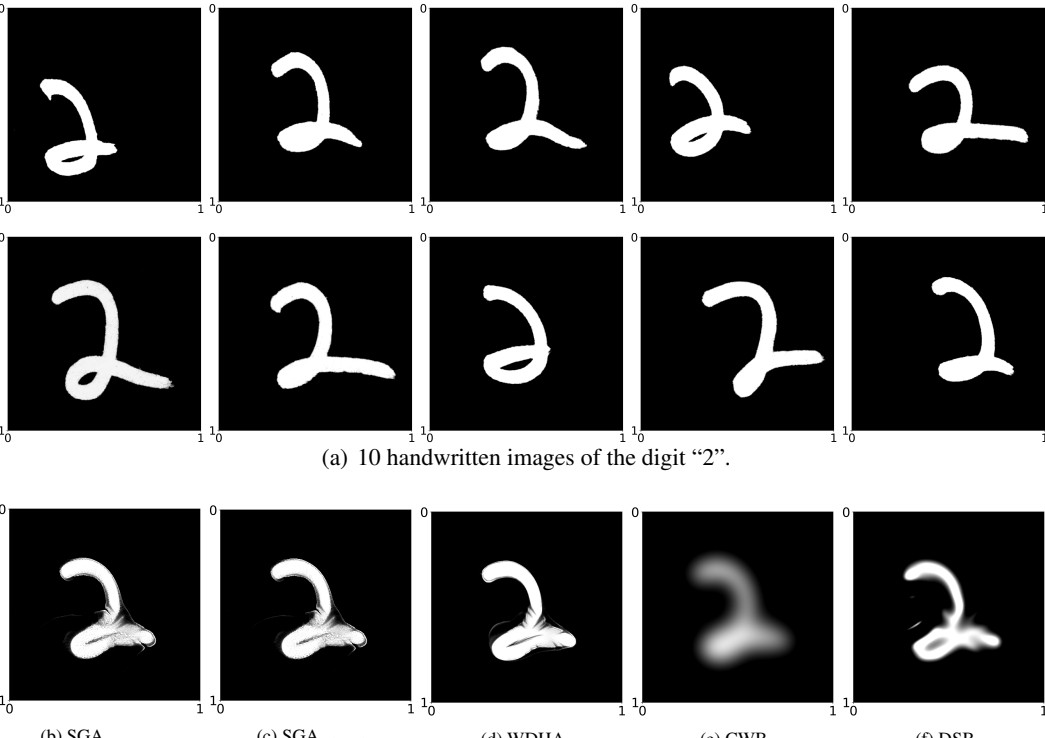

(a) 10 handwritten images of the digit "2".

(b) $\text{SGA}_{\eta_t=0.1}$     (c) $\text{SGA}_{\text{Adagrad}}$     (d) WDHA     (e) CWB     (f) DSB

Figure 6: Input handwritten digits (top) and barycenters computed by different algorithms (bottom).

| Method | Barycenter Functional Value [↓] | Time [↓] |
|---|---|---|
| SGA | $5.992 \times 10^{-3}$ | **406** |
| SGA(Adagrad) | $\mathbf{5.992 \times 10^{-3}}$ | 409 |
| WDHA | $6.049 \times 10^{-3}$ | 415 |
| CWB | $6.463 \times 10^{-3}$ | 430 |
| DSB | $6.021 \times 10^{-3}$ | 670 |

Table 4: Comparison of barycenter functional value and computation time for different algorithms.

### B.4 GAUSSIAN DISTRIBUTIONS

We consider three Gaussian probability measures $\mu_1, \mu_2, \mu_3$ on $\mathbb{R}^2$, where $\mu_i \sim \mathcal{N}(m_i, \Sigma_i), i = 1, \ldots, 3$. Here, we set $m_1 = (40, 40)^T, m_2 = (50, 50)^T, m_3 = (20, 50)^T$, and

$$\Sigma_1 = \begin{pmatrix} 0.2^2 & 0 \\ 0 & 0.2^2 \end{pmatrix}, \quad \Sigma_2 = \begin{pmatrix} 0.2^2 & 0 \\ 0 & 0.6^2 \end{pmatrix}, \quad \Sigma_3 = \begin{pmatrix} 0.4^2 & 0 \\ 0 & 0.4^2 \end{pmatrix}.$$

We note that our methods work on a compact domain, thus those distribution are truncated on a large enough square, i.e., $[0, 100]^2$, to make the truncation error sufficiently small. The Wasserstein barycenter $\bar{\mu}$ of $\mu_1, \mu_2, \mu_3$ is again Gaussian, $\bar{\mu} \sim \mathcal{N}(\bar{m}, \bar{\Sigma})$, with barycenter mean and standard deviation given by

$$\bar{m} = \left( \frac{110}{3}, \frac{140}{3} \right)^\top \text{ and } \bar{\Sigma} = \begin{pmatrix} \left(\frac{4}{15}\right)^2 & 0 \\ 0 & 0.4^2 \end{pmatrix}.$$

We set $\eta = 0.005$ for SGA and WDHA, reg $= 0.005$ for CWB and DSB. We represent all truncated Gaussian distributions on an equally spaced $1024 \times 1024$ grid, and evaluate the squared 2-Wasserstein distance between the computed barycenter and $\bar{\mu}$ using the Back-and-Forth approach (Jacobs et al., 2021). In this Gaussian setup, our method(SGA) returns the smallest squared 2-Wasserstein distance between computed barycenter and the groundtruth barycenter $\bar{\mu}$ as shown in Table 5. The barycenter fucntional value is shown in Table 6.

| SGA | WDHA | CWB | DSB |
|---|---|---|---|
| **0.4636** | 20.3449 | 21.7867 | 0.4925 |

Table 5: Squared 2-Wasserstein distance between the $\bar{\mu}$ and the barycenter computed by each method.

| Groundtruth | SGA | WDHA | CWB | DSB |
|---|---|---|---|---|
| 88.9067 | 89.0074 | 102.0016 | 109.4954 | 89.0247 |

Table 6: Barycenter functional value for groundtruth and different methods.

We compare the approximation error of the optimal transport maps from the barycenter to each distribution for our method and for WDHA, which also relies on Kantorovich potentials to compute the Wasserstein barycenter. We note that CWB and DSB are entopic regularized methods and does not produce optimal transport maps without further modifications. The approximate optimal transport maps $\hat{T}_i$ are obtained from the learned Kantorovich potentials $\hat{f}_i$ via $\hat{T}_i(x) = x - \nabla \hat{f}_i(x), i = 1, 2$ and $\hat{T}_3(x) = x - \nabla \hat{f}_{\mathrm{mix}}(x)$, with $\hat{f}_{\mathrm{mix}} = -\hat{f}_1 - \hat{f}_2$. We evaluate the approximation error between the learned maps $\hat{T}_i$ and the ground-truth optimal transport maps $T_i$ from $\bar{\mu}$ to $\mu_i$, $T_i(x) := m_i + \bar{\Sigma}^{-1/2}(\bar{\Sigma}^{1/2}\Sigma_i\bar{\Sigma}^{1/2})^{1/2}\bar{\Sigma}^{-1/2}(x - \bar{m})$, using the relative $L^2$ norm $\|T_i - \hat{T}_i\|_2^2 / \|T_i\|_2^2$, where $\|T_i - \hat{T}_i\|_2^2 := \int_\Omega \|T_i(x) - \hat{T}_i(x)\|_2^2 \, dx$. The results are provided in the following Table 7. As shown in Table 7, the approximation errors for $\hat{T}_1$, $\hat{T}_2$, and $\hat{T}_3$ under our method are smaller than those obtained with WDHA.

| | SGA | WDHA |
|---|---|---|
| $\|T_1 - \hat{T}_1\|_2^2 / \|T_1\|_2^2$ | $\mathbf{6.6620 \times 10^{-2}}$ | $6.7319 \times 10^{-2}$ |
| $\|T_2 - \hat{T}_2\|_2^2 / \|T_2\|_2^2$ | $\mathbf{4.1357 \times 10^{-2}}$ | $4.2259 \times 10^{-2}$ |
| $\|T_3 - \hat{T}_3\|_2^2 / \|T_3\|_2^2$ | $\mathbf{4.0582 \times 10^{-2}}$ | $4.0961 \times 10^{-2}$ |

Table 7: Comparison of approximation errors for the optimal transport maps learned by SGA and WDHA. Smaller values (in bold) indicate better approximation.

## C  THE USE OF LARGE LANGUAGE MODELS (LLMs)

A Large Language Model (LLM) was used solely to identify grammar mistakes, polish language, and detect typographical errors. No part of the research ideation, discovery, or substantive content was generated by the LLM.

