# OpenReview forum: "Sobolev Gradient Ascent for Optimal Transport: Barycenter Optimization and Convergence Analysis"
_ICLR.cc/2026/Conference — ICLR 2026 Poster_

### Official Review · Reviewer_h7bp · 2025-10-31

**Soundness:** 3
**Presentation:** 3
**Contribution:** 2
**Rating:** 4
**Confidence:** 3

**Summary:**

In this paper,  the authors propose an algorithm for computation of the exact Wasserstein barycenter for a collection of input distributions discretized over a regular grid. The algorithm is based on the idea of Sobolev gradient descent and is tested in several experimental setups.

**Strengths:**

In general, the paper is well-written and provides an algorithm supported with theoretical analysis and experimental illustrations in several setups.

**Weaknesses:**

My main concerns are related to the limited practical usefulness of the developed algorithm. For example, as the authors claim themself in lines 101-105, the developed algorithm is not suitable for the distributions of the dimensions higher than 3D. Besides, the algorithm is designed for the computation of the optimal transport barycenter under the quadratic cost function assumption. This point also could be listed among the limitations of the developed methodology. While I understand that the listed limitations might correspond not to the algorithm itself, but rather to the particular  class of the algorithms for exact barycenter computation, I have other concerns regarding the demonstration of the practicality of the proposed algorithm.

Specifically, all of the experimental results only show the ability of the method to compute the barycenter of distributions, but do not assess its ability to approximate well the optimal transport (OT) maps between each of the distributions and barycenter. I think that it is an important aspect since the learned maps have more practical use cases than the barycenter itself. Meanwhile, the authors do not perform comparison with the ground-truth barycenters which are known, e.g., for Gaussian distributions. Thus, I kindly suggest the authors to perform comparison of their approach with the ground-truth barycenter and OT maps in the setting with Gaussian distributions which are known thanks to (Chewi et al., 2020).

The related work section lacks the overview of recent methods for approximating the barycenter of continuous distributions (Fan et al, 2021; Chi et al., 2023; Noble et al., 2023; Kolesov et al, 2024a,b). I think these algorithms deserve mentioning since in lines 106-107 you write that “almost all existing barycenter algorithms are limited to 2D problems” which is not true when we are talking about the methods listed above. Thus, I kindly suggest the authors add the references in the related work sections and make the phrase in lines 106-107 more clear.

**Overall**, I think that the experimental evaluation of the proposed approach  should be improved since now it lacks comparison with the ground-truth barycenters and OT maps which is important for clearly revealing the performance of the developed approach.

**References.**

Fan, J., Taghvaei, A., and Chen, Y. Scalable computations of wasserstein barycenter via input convex neu- ral networks. In Meila, M. and Zhang, T. (eds.), Pro- ceedings of the 38th International Conference on Ma- chine Learning, volume 139 of Proceedings of Machine Learning Research, pp. 1571–1581. PMLR, 18–24 Jul 2021.

Chi, J., Yang, Z., Li, X., Ouyang, J., and Guan, R. Varia- tional wasserstein barycenters with c-cyclical monotonic- ity regularization. In Proceedings of the AAAI Confer- ence on Artificial Intelligence, volume 37, pp. 7157–7165, 2023.

Noble, M., Bortoli, V. D., Doucet, A., and Durmus, A. Tree-based diffusion schro ̈dinger bridge with applica- tions to wasserstein barycenters. In Thirty-seventh Conference on Neural Information Processing Systems, 2023.

Kolesov, A., Mokrov, P., Udovichenko, I., Gazdieva, M., Pammer, G., Burnaev, E., and Korotin, A. Energy-guided continuous entropic barycenter estimation for general costs. Advances in Neural Information Processing Systems, 2024a

Kolesov, A., Mokrov, P., Udovichenko, I., Gazdieva, M., Pammer, G., Burnaev, E., and Korotin, A. Estimating barycenters of distributions with neural optimal transport. In Forty-first International Conference on Machine Learning, 2024b

Sinho Chewi, Tyler Maunu, Philippe Rigollet, and Austin J. Stromme. Gradient descent algorithms for Bures-Wasserstein barycenters. In Jacob Abernethy and Shivani Agarwal (eds.), Proceedings of Thirty Third Conference on Learning Theory, volume 125 of Proceedings of Machine Learning Research, pp. 1276–1304. PMLR, 09–12 Jul 2020.

**Questions:**

- Could you perform comparison with the ground-truth barycenter and OT maps in the Gaussian setup?
- Could your approach be extended to the case of more general cost functions?

---

> ### Author Response · Authors · 2025-11-22
> **Response to Reviewer h7bp**
>
> We appreciate your constructive comments!
> > My main concerns are related to the limited practical usefulness of the developed algorithm. For example, as the authors claim themself in lines 101-105, the developed algorithm is not suitable for the distributions of the dimensions higher than 3D.
>
> To the best of our knowledge, all existing high-accuracy exact barycenter algorithms, which are the focus of this work, are limited to two or three dimensions. Widely used libraries such as POT only support entropic barycenter solvers for one or two dimensional distributions. For higher dimensional settings, the existing literature either relies on neural network based approaches [1], [2] or considers a regularized notion of barycenter [3], both of which trade numerical accuracy for scalability.
>
> >Besides, the algorithm is designed for the computation of the optimal transport barycenter under the quadratic cost function assumption. This point also could be listed among the limitations of the developed methodology. Could your approach be extended to the case of more general cost functions?
>
> Our method applies to cost functions of the form $c(x,y)=h(x-y)$, where $h$ is strictly convex and even. Under this assumption, the Sobolev gradient and its computation are well defined, see Proposition 2.9 in [4] for the differentiability and section 4.1 in [5] for computational details. However, in Euclidean space, the sample mean minimizes the sum of squared $L^2$-distances, whereas under the $L^1$-distance the minimizer becomes the sample median. Analogously, the Wasserstein barycenter is defined as the minimizer of the sum of squared 2-Wasserstein distances [6].

---

> ### Author Response · Authors · 2025-11-22
> **Response to Reviewer h7bp**
>
> > Specifically, all of the experimental results only show the ability of the method to compute the barycenter of distributions, but do not assess its ability to approximate well the optimal transport (OT) maps between each of the distributions and barycenter. I think that it is an important aspect since the learned maps have more practical use cases than the barycenter itself. Meanwhile, the authors do not perform comparison with the ground-truth barycenters which are known, e.g., for Gaussian distributions. Thus, I kindly suggest the authors to perform comparison of their approach with the ground-truth barycenter and OT maps in the setting with Gaussian distributions which are known thanks to (Chewi et al., 2020). Could you perform comparison with the ground-truth barycenter and OT maps in the Gaussian setup?
>
> Sure thing. We consider three Gaussian probability measures $\mu_1,\mu_2,\mu_3$ on $\mathbb{R}^2$, where $\mu_i \sim \mathcal{N}(m_i, \Sigma_i), i = 1,\dots,3$. Here, we set $m_1 = (40, 40)^T,
> m_2 = (50, 50)^T,
> m_3 = (20, 50)^T,$
> and $\Sigma_1 = \operatorname{diag}(0.2^2, 0.2^2)$,
> $\Sigma_2 = \operatorname{diag}(0.2^2, 0.6^2)$,
> $\Sigma_3 = \operatorname{diag}(0.4^2, 0.4^2).$
> We note that our methods work on a compact domain, thus those distribution are truncated on a large enough square, i.e.,  $[0,100]^2$, to make the truncation error sufficiently small. The Wasserstein barycenter $\bar\mu$ of $\mu_1, \mu_2, \mu_3$ is again Gaussian, $ \bar\mu \sim \mathcal{N}(\bar m, \bar\Sigma)$, with barycenter mean and standard deviation given by
> $\bar m = \left(\frac{110}{3},\frac{140}{3}\right)^\top$ and $\bar\Sigma =\operatorname{diag}(\left(\frac{4}{15}\right)^2, 0.4^2).$
>
> We set $\eta = 0.005$ for SGA and WDHA, $\text{reg} = 0.005$ for CWB and DSB. We represent all truncated Gaussian distributions on an equally spaced $1024 \times 1024$ grid, and evaluate the squared 2-Wasserstein distance between the computed barycenter and $\bar\mu$ using the Back-and-Forth approach [5]. In this Gaussian setup, our method(SGA) returns the smallest squared 2-Wasserstein distance between computed barycenter and the groundtruth barycenter $\bar\mu$ as shown in Table5. The barycenter functional value is shown in Table 6. (Table 5 and 6 in the revised manuscript are pasted below for your convenience.)
>
> *Table 5: Squared 2-Wasserstein distance between $\bar\mu$ and the barycenter computed by each method.*
>
> | SGA      | WDHA    | CWB     | DSB     |
> |----------|---------|---------|---------|
> | **0.4636** | 20.3449 | 21.7867 | 0.4925  |
>
> *Table 6: Barycenter functional value for groundtruth and different methods.*
>
> | Groundtruth |   SGA   |   WDHA   |   CWB   |   DSB   |
> |------------:|--------:|---------:|--------:|--------:|
> | 88.9067     | 89.0074 | 102.0016 | 109.4954 | 89.0247 |
>
>
>
> We evaluate the approximation error between the learned maps $\hat T_i$ and the ground-truth optimal transport maps $T_i$ from $\bar\mu$ to $\mu_i$, $T_i(x) := m_i + \bar\Sigma^{-1/2}\left(\bar\Sigma^{1/2}\Sigma_i\bar\Sigma^{1/2}\right)^{1/2}\bar\Sigma^{-1/2}\,(x - \bar m)$, using the relative $L^2$ norm $\Vert T_i - \hat T_i\Vert_2^2 / \Vert T_i\Vert_2^2$, where $\lVert T_i - \hat T_i \rVert_2^2 := \int_{\Omega} \lVert T_i(x) - \hat T_i(x) \rVert_2^2 \, dx$. The results are provided in the following Table 7. (Table 7 in the revised manuscript is pasted below for your convenience.)  As shown in Table 7, the approximation errors for $\hat T_1$, $\hat T_2$, and $\hat T_3$ under our method are smaller than those obtained with WDHA.
>
> *Table 7: Comparison of approximation errors for the optimal transport maps learned by SGA and WDHA. Smaller values (in bold) indicate better approximation.*
> |                                   | SGA                              | WDHA                             |
> |-----------------------------------|----------------------------------|----------------------------------|
> | $\Vert T_1 - \hat T_1\Vert_2^2 / \Vert T_1\Vert_2^2$ | **$\mathbf{6.6620\times10^{-2}}$**        | $6.7319\times10^{-2}$            |
> | $\Vert T_2 - \hat T_2\Vert_2^2 / \Vert T_2\Vert_2^2$ | **$\mathbf{4.1357\times10^{-2}}$**        | $4.2259\times10^{-2}$            |
> | $\Vert T_3 - \hat T_3\Vert_2^2 / \Vert T_3\Vert_2^2$ | **$\mathbf{4.0582\times10^{-2}}$**        | $4.0961\times10^{-2}$            |
>
>
>
>
> This part of results are included in Subsection B.4 of appendix in the revised manuscript.

---

> > ### Author Response · Authors · 2025-11-22
> > **Response to Reviewer h7bp**
> >
> > >  The related work section lacks the overview of recent methods for approximating the barycenter of continuous distributions ([1]; [2]; [3]; [7]). I think these algorithms deserve mentioning since in lines 106-107 you write that “almost all existing barycenter algorithms are limited to 2D problems” which is not true when we are talking about the methods listed above. Thus, I kindly suggest the authors add the references in the related work sections and make the phrase in lines 106-107 more clear.
> >
> > Thank you for pointing out those references. We revised the content in between lines 106 and 107 to ``However, to the best of our knowledge, all existing high-accuracy exact barycenter algorithms are limited to 2D or 3D problems. The entropic methods implemented in the POT library [8] are limited to the computation of barycenters for 2D distributions. For higher dimensional problems, existing literature either rely on neural network based methods [1], [2] or target a regularized notion of barycenter [3], which trade numerical accuracy for scalability."
> >
> > In addition, we cited the following regularized approaches on line 81:
> > - Kolesov, A., Mokrov, P., Udovichenko, I., Gazdieva, M., Pammer, G., Burnaev, E., and Korotin, A. *Energy-guided continuous entropic barycenter estimation for general costs*. In *Advances in Neural Information Processing Systems*, 2024a.
> >
> > - Noble, M., Bortoli, V. D., Doucet, A., and Durmus, A. *Tree-based diffusion Schrödinger bridge with applications to Wasserstein barycenters*. In *Thirty-seventh Conference on Neural Information Processing Systems*, 2023.
> >
> > - Chi, J., Yang, Z., Li, X., Ouyang, J., and Guan, R. *Variational Wasserstein barycenters with c-cyclical monotonicity regularization*. In *Proceedings of the AAAI Conference on Artificial Intelligence*, 37:7157–7165, 2023.
> >
> > [1] Jiaojiao Fan, Amirhossein Taghvaei, and Yongxin Chen. Scalable computations of wasserstein barycenter via input convex neural networks. In Proceedings of the 42nd International Conference on Machine Learning (ICML), Proceedings of Machine Learning Research, 2025.
> >
> > [2] Alexander Kolesov, Petr Mokrov, Igor Udovichenko, Milena Gazdieva, Gudmund Pammer, Evgeny Burnaev, and Alexander Korotin. Estimating barycenters of distributions with neural optimal transport. In Proceedings of the 41st International Conference on Machine Learning, volume 235 of Proceedings of Machine Learning Research, pp. 25016–25041. PMLR, 21–27
> > Jul 2024.
> >
> > [3] Jinjin Chi, Zhiyao Yang, Ximing Li, Jihong Ouyang, and Renchu Guan. Variational wasserstein barycenters with c-cyclical monotonicity regularization. In Proceedings of the AAAI Conference on Artificial Intelligence, volume 37, pp.7157–7165, 2023.
> >
> > [4] Wilfrid Gangbo. An Introduction to the Mass Transportation Theory and its Applications, 2004. URL https://www.math.ucla.edu/~wgangbo/publications/notecmu.pdf.
> >
> > [5] Matt Jacobs, Wonjun Lee, and Flavien Léger. The back-and-forth method for
> > wasserstein gradient flows. ESAIM: Control, Optimisation and Calculus of
> > Variations, 27:28, 2021.
> >
> > [6] M. Agueh and G. Carlier. Barycenters in the Wasserstein space. SIAM J. Math.
> > Anal., 43(2):904–924, 2011. ISSN 0036-1410. doi: 10.1137/100805741.
> >
> > [7] Noble, M., Bortoli, V. D., Doucet, A., and Durmus, A. Tree-based dif-
> > fusion schrodinger bridge with applications to wasserstein barycenters.
> > In Thirty-seventh Conference on Neural Information Processing Systems,
> > 2023.
> >
> > [8] Rémi Flamary, Nicolas Courty, Alexandre Gramfort, Mokhtar Z. Alaya, Aurélie
> > Boisbunon, Stanislas Chambon, Laetitia Chapel, Adrien Corenflos, Kilian
> > Fatras, Nemo Fournier, Léo Gautheron, Nathalie T.H. Gayraud, Hicham Ja-
> > nati, Alain Rakotomamonjy, Ievgen Redko, Antoine Rolet, Antony Schutz,
> > Vivien Seguy, Danica J. Sutherland, Romain Tavenard, Alexander Tong, and
> > Titouan Vayer. POT: Python optimal transport. Journal of Machine Learning
> > Research, 22(78):1–8, 2021. URL http://jmlr.org/papers/v22/20-451.
> > html.

---

### Official Review · Reviewer_Vdfa · 2025-10-31

**Soundness:** 4
**Presentation:** 4
**Contribution:** 3
**Rating:** 8
**Confidence:** 3

**Summary:**

This paper introduces a novel, constraint-free, and concave dual formulation for the Wasserstein barycenter problem. By removing the constraints present in previous dual formulations, the authors develop a scalable Sobolev Gradient Ascent (SGA) algorithm. A key theoretical contribution is a global convergence analysis for the proposed method. The efficacy of the algorithm is validated through numerical experiments that show superior performance over existing methods.

**Strengths:**

The paper focuses on a fundamental problem of Barycenter Optimization. Theorem 4 is the paper's cornerstone. It introduces a constraint-free concave dual formulation for the Wasserstein barycenter problem. This key result makes the problem amenable to optimization via SGA and allows the authors to establish a global convergence guarantee.

**Weaknesses:**

1. The method's efficiency hinges on a regular grid, which allows for an efficient FFT-based solver for the inverse Laplacian. It would significantly strengthen the paper to discuss the adaptations required for irregular meshes, where one would need to employ more computationally expensive methods.
2. The grid-based approach is subject to the curse of dimensionality, meaning its computational cost grows exponentially with the number of dimensions. To their credit, the authors explicitly acknowledge this and correctly position their method for low-dimensional (2D and 3D) problems.

**Questions:**

Please fix a hanging citation "Matthew Jacobs and Bohan Zhou. The Signed Wasserstein Barycenters, 2025". It does not seem to exist?

---

> ### Author Response · Authors · 2025-11-22
> **Response to Reviewer Vdfa**
>
> We appreciate your constructive comments!
>
> >  The method's efficiency hinges on a regular grid, which allows for an efficient FFT-based solver for the inverse Laplacian. It would significantly strengthen the paper to discuss the adaptations required for irregular meshes, where one would need to employ more computationally expensive methods.
>
> To extended our approach from regular to non-uniform grids, three components require modification. First, the fast $c$-transform must be replaced, for example, by adapting the parametric Legendre transform algorithm by [1]. Second, the FFT-based Poisson solver can be substituted by multigrid methods. Third, the computation of the pushforward measure, currently obtained by pushing forward discretized marginals under a discretized map, would also need multigrid techniques to handle the Jacobian. Since multigrid methods are substantially more expensive than FFT-based Poisson solvers, higher computational costs are expected. This part of discussion is added to section 1.2 of the main paper.
>
> >The grid-based approach is subject to the curse of dimensionality, meaning its computational cost grows exponentially with the number of dimensions. To their credit, the authors explicitly acknowledge this and correctly position their method for low-dimensional (2D and 3D) problems.
>
> To the best of our knowledge, all existing high-accuracy exact barycenter algorithms, which are the focus of this work, are limited to two or three dimensions. Widely used libraries such as POT only support entropic barycenter solvers for one or two dimensional distributions. For higher dimensional settings, the existing literature either relies on neural network based approaches ([2], [3]) or considers a regularized notion of barycenter [4], both of which trade numerical accuracy for scalability.
>
> > Please fix a hanging citation "Matthew Jacobs and Bohan Zhou. The Signed Wasserstein Barycenters, 2025". It does not seem to exist?
>
> Fixed, thank you.
>
>
> [1] J Hiriart-Urruty and Yves Lucet. Parametric computation of the legendre fenchel conjugate with application to the computation of the moreau envelope. Journal of Convex Analysis, 14(3):657, 2007.
>
> [2] Jiaojiao Fan, Amirhossein Taghvaei, and Yongxin Chen. Scalable computations of wasserstein barycenter via input convex neural networks. In Proceedings of the 42nd International Conference on Machine Learning (ICML), Proceedings of Machine Learning Research, 2025.
>
> [3] Alexander Kolesov, Petr Mokrov, Igor Udovichenko, Milena Gazdieva, Gudmund Pammer, Evgeny Burnaev, and Alexander Korotin. Estimating barycenters of distributions with neural optimal transport. In Proceedings of the 41st International Conference on Machine Learning, volume 235 of Proceedings of Machine Learning Research, pp. 25016–25041. PMLR, 21–27
> Jul 2024.
>
> [4] Jinjin Chi, Zhiyao Yang, Ximing Li, Jihong Ouyang, and Renchu Guan. Variational wasserstein barycenters with c-cyclical monotonicity regularization. In Proceedings of the AAAI Conference on Artificial Intelligence, volume 37, pp.7157–7165, 2023.

---

### Official Review · Reviewer_VvTD · 2025-11-01

**Soundness:** 2
**Presentation:** 3
**Contribution:** 3
**Rating:** 6
**Confidence:** 2

**Summary:**

This paper proposes a constraint-free concave dual formulation for the Wasserstein barycenter and a dual ascent algorithm adapted to the Sobolev geometry. The algorithm matches the complexity rate of the subgradient method. A key advantage is that the new dual algorithm bypasses the expensive c-concavity projection and achieves high efficiency. The experiments demonstrate the advantages of the proposed method.

**Strengths:**

This paper presents a novel concave dual formulation for the Wasserstein barycenter problem that is unconstrained yet achieves strong duality. The theoretical analysis appears reasonable and expected. Empirically, the authors demonstrate that SGA is significantly faster and more numerically stable.

**Weaknesses:**

The stepsize rule seems impractical—it requires an M that depends on all iterates. It's somewhat bizarre that the stepsize depends on future iterates. If the stepsize can be chosen more freely, please provide a detailed convergence rate analysis.

Additionally, would an adaptive gradient method (such as AdaGrad) perform well in your examples?

As the authors pointed out, the algorithm's complexity scales with the grid size $n$, which scales *exponentially* with the dimension $d$. This effectively limits the current method to low-dimensional problems ($d=2, 3$). This could be a practical constraint for many machine learning applications.

The real-world example using electric scooter tracking data appears quite simple. Are there more challenging datasets in this area?

**Questions:**

See the weakness part

**Details Of Ethics Concerns:**

Not available..

---

> ### Author Response · Authors · 2025-11-22
> **Response to Reviewer VvTD**
>
> We appreciate your constructive comments!
> > The stepsize rule seems impractical—it requires an M that depends on all iterates. It's somewhat bizarre that the stepsize depends on future iterates. If the stepsize can be chosen more freely, please provide a detailed convergence rate analysis. Additionally, would an adaptive gradient method (such as AdaGrad) perform well in your examples?
>
> Assuming boundedness of $M$, any step-size choice that satisfies $\sum_{t=1}^T \eta_t = \infty$ and $\sum_{t=1}^{T} \eta_t^2 < \infty$ would lead to the $O(1/\sqrt{T})$ rate. This matches with the standard convergence rate for gradient descent of convex-nonsmooth functions in Euclidean space. The option in the main paper is a convenient choice. We can adapt the AdaGrad scheme to our barycenter computation algorithm (yielding SGA–AdaGrad), described in the following.
>
> **Algorithm : AdaGrad-adapted SGA for barycenter computation**
>
> Require: step size $\eta > 0$, small constant $\varepsilon > 0$.
>
> Initialize: $f_i^{(0)}$ and $r_i^{(0)} \leftarrow 0$ for $i = 1,\dots,m-1$.
>
> For $t = 1,\dots,T$:
> &nbsp;&nbsp;For $i = 1,\dots,m-1$:
> &nbsp;&nbsp;&nbsp;&nbsp;$g_i^{(t)} \leftarrow \nabla_{f_i} \mathcal{D}\bigl(f_1^{(t-1)}, \ldots, f_{m-1}^{(t-1)}\bigr)$
> &nbsp;&nbsp;&nbsp;&nbsp;$r_i^{(t)} \leftarrow r_i^{(t-1)} + g_i^{(t)} \odot g_i^{(t)}$
> &nbsp;&nbsp;&nbsp;&nbsp;$f_i^{(t)} \leftarrow f_i^{(t-1)} + \eta \,\dfrac{g_i^{(t)}}{\sqrt{r_i^{(t)}} + \varepsilon}$
>
>
> We applied SGA–AdaGrad to the newly added handwritten data example. The results, presented in Appendix B3, show that it performs similarly to SGA with a constant step size, with only minor differences in both the barycenter functional value and the computation time. (Table 4 in Appendix B3 of the revised manuscript is pasted below for your convenience.) This is expected, as when the objective is convex and nonsmooth, both methods typically exhibit comparable behavior.
>
> *Table 4: Comparison of barycenter functional value and computation time for different algorithms.*
>
> | Method       | Barycenter Functional Value [↓] | Time [↓] |
> |-------------|----------------------------------|---------:|
> | SGA         | $5.9922\times10^{-3}$            | **406**  |
> | SGA (AdaGrad) | $\mathbf{5.992\times10^{-3}}$  | 409      |
> | WDHA        | $6.0487\times10^{-3}$            | 415      |
> | CWB         | $6.4626\times10^{-3}$            | 430      |
> | DSB         | $6.0212\times10^{-3}$            | 670      |
>
>
> >  As the authors pointed out, the algorithm's complexity scales with the grid size , which scales exponentially with the dimension. This effectively limits the current method to low-dimensional problems (d=2,3). This could be a practical constraint for many machine learning applications.
>
> To overcome the curse of dimensionality in optimal transport barycenter problems, existing approaches typically rely on deep-learning–based methods [1], which trade numerical accuracy for scalability. The investigation of scalable Sobolev-gradient–based approaches with neural networks is left for future work. To the best of our knowledge, no high-accuracy algorithm is currently available for the exact barycenter when dimension is higher than three. Even widely used libraries such as POT only support entropic barycenter solvers for one- or two-dimensional distributions.

---

> ### Author Response · Authors · 2025-11-22
> **Response to Reviewer VvTD**
>
> > The real-world example using electric scooter tracking data appears quite simple. Are there more challenging datasets in this area?
>
> The handwritten digits data is commonly employed to access performance of barycenter algorithms, see [2],[3],[4] among others. We compare the proposed methods on high-resolution handwritten digit data from [5], treating each image as a probability density supported on $[0,1]^2$. Specifically, we compute the Wasserstein barycenter of 10 images of the digit 2, each of size $500 \times 500$ pixels. For SGA and WDHA, we use a constant step size $\eta_t = 0.1$, while for SGA-Adagrad we set $\eta = 0.1$ and $\epsilon = 10^{-8}$. The regularization parameters for CWB and DSB are both chosen as $5 \times 10^{-3}$. The results are shown in Table 4 and Figure 6 of Appendix, Subsection B3. Among all methods, SGA produces the barycenter with the finest visible texture, while also achieving the lowest barycenter functional value and the shortest computation time.
>
> [1] Jiaojiao Fan, Amirhossein Taghvaei, and Yongxin Chen. Scalable computations of Wasserstein barycenter via input convex neural networks. In Proceedings of the 42nd International Conference on Machine Learning (ICML), Proceedings
> of Machine Learning Research, 2025.
>
> [2] Marco Cuturi and Arnaud Doucet. Fast computation of Wasserstein barycenters. In Proceedings of the 31st International Conference on Machine Learning, pp. 685–693, 2014.
>
> [3] Dongdong Ge, Haoyue Wang, Zikai Xiong, and Yinyu Ye. Interior-point methods strike back: Solving the wasserstein barycenter problem. Advances in neural information processing systems, 32, 2019.
>
> [4] Kaheon Kim, Rentian Yao, Changbo Zhu, and Xiaohui Chen. Optimal transport barycenter via nonconvex concave minimax optimization. In International Conference on Machine Learning (ICML), July 2025.
>
> [5] Cédric Beaulac and Jeffrey S Rosenthal. Introducing a new high-resolution handwritten digits data set with writer characteristics. SN Computer Science, 4(1):66, 2022.

---

### Official Review · Reviewer_DYXD · 2025-11-02

**Soundness:** 3
**Presentation:** 3
**Contribution:** 2
**Rating:** 6
**Confidence:** 3

**Summary:**

This paper proposed a constraint-free and concave formulation of the Wasserstein barycenter optimization problem which achieves strong duality.  The paper derived a scalable Sobolev gradient ascent (SGA) algorithm without the computationally expensive c-concave projection steps. Then, it showed that the proposed SGA algorithm achieves the same rate as the classical subgradient descent methods for minimizing nonsmooth convex functions in the Euclidean space. Numerical experiments demonstrate the effectiveness of SGA over the existing optimal transport barycenter solvers.

**Strengths:**

Overall I found this paper to be interesting and provided a good theoretical contribution in computing Wasserstein barycenter efficiently:
-  the constraint-free concave dual formulation allows the algorithm to avoid the OT map computations in the primal problem
-  the proposed Sobolev gradient ascent (SGA) algorithm is simple and straightforward, but matches the rate of classical subgradient descent for minimizing nonsmooth convex functions in the Euclidean space.
- with synthetic and real world data, it was shown that SGA achieved comparable or better performance than existing baselines.

**Weaknesses:**

- In comparison to the theoretical results, I found the numerical experiments to be on the weaker side, with limited data involved in testing SGA. For real-world data, only one set of video frames was used to demonstrate SGA's performance, and the comparison between WDHA and SGA's performance was not very obvious. WDHA also captured the trajectory of the moving object, but for some reason the picture appeared to be darker. The paper will be strengthened if the real-world data experiments part can be more comprehensive and quantitative.

**Questions:**

Is the convergence rate for SGA in theorem 6 optimal?

---

> ### Author Response · Authors · 2025-11-22
> **Response to Reviewer DYXD**
>
> We appreciate your constructive comments!
> > In comparison to the theoretical results, I found the numerical experiments to be on the weaker side, with limited data involved in testing SGA. For real-world data, only one set of video frames was used to demonstrate SGA's performance, and the comparison between WDHA and SGA's performance was not very obvious. WDHA also captured the trajectory of the moving object, but for some reason the picture appeared to be darker. The paper will be strengthened if the real-world data experiments part can be more comprehensive and quantitative.
>
> Thank you, we added an additional handwritten digits data example, which is commonly employed to access performance of barycenter algorithms, see [1]; [2]; [3] among others.  among others. We compare the proposed methods on high-resolution handwritten digit data from [4], treating each image as a probability density supported on $[0,1]^2$. Specifically, we compute the Wasserstein barycenter of 10 images of the digit 2, each of size $500 \times 500$ pixels. For SGA and WDHA, we use a constant step size $\eta_t = 0.1$, while for SGA-Adagrad we set $\eta = 0.1$ and $\epsilon = 10^{-8}$. The regularization parameters for CWB and DSB are both chosen as $5 \times 10^{-3}$. The results are shown in Table 4 and Figure 6 of Appendix, Subsection B3. (Table 4 is pasted below for your convenience.) Among all methods, SGA produces the barycenter with the finest visible texture, while also achieving the lowest barycenter functional value and the shortest computation time. WDHA appears darker, likely due to its inferior performance. The corresponding barycenter functional values are:
> SGA ($3.11\times 10^{-5}$), WDHA ($3.675\times 10^{-5}$),
> CWB ($3.223\times 10^{-5}$), and DSB ($3.206\times 10^{-5}$).
>
> *Table 4: Comparison of barycenter functional value and computation time for different algorithms.*
>
> | Method       | Barycenter Functional Value [↓] | Time [↓] |
> |-------------|----------------------------------|---------:|
> | SGA         | $5.9922\times10^{-3}$            | **406**  |
> | SGA (AdaGrad) | $\mathbf{5.992\times10^{-3}}$  | 409      |
> | WDHA        | $6.0487\times10^{-3}$            | 415      |
> | CWB         | $6.4626\times10^{-3}$            | 430      |
> | DSB         | $6.0212\times10^{-3}$            | 670      |
>
>
>
> > Is the convergence rate for SGA in theorem 6 optimal?
>
> The $O(1/\sqrt{T})$ rate in our paper is optimal because the objective function is convex but not $L$-smooth. In this setting, the rate matches the minimax lower bound established in Theorems 3.2.1 and 3.2.2 of [5].
>
> [1] Marco Cuturi and Arnaud Doucet. Fast computation of Wasserstein barycenters. In Proceedings of the 31st International Conference on Machine Learning, pp. 685–693, 2014.
>
> [2] Dongdong Ge, Haoyue Wang, Zikai Xiong, and Yinyu Ye. Interior-point methods strike back: Solving the wasserstein barycenter problem. Advances in neural information processing systems, 32, 2019.
>
> [3] Kaheon Kim, Rentian Yao, Changbo Zhu, and Xiaohui Chen. Optimal transport barycenter via nonconvex concave minimax optimization. In International Conference on Machine Learning (ICML), July 2025.
>
> [4] Cédric Beaulac and Jeffrey S Rosenthal. Introducing a new high-resolution handwritten digits data set with writer characteristics. SN Computer Science, 4(1):66, 2022.
>
> [5] Yurii Nesterov. Introductory lectures on convex optimization: A basic course, volume 87. Springer Science & Business Media, 2013.

---

### Author Response · Authors · 2025-11-29
**Summary**

In this paper, we introduce a concave formulation of the exact Wasserstein barycenter problem and develop a Sobolev-gradient algorithm (SGA) for its computation. Our method enjoys global convergence guarantees and efficiently produces highly accurate barycenter densities for 2D and 3D input distributions discretized on regular grids. As demonstrated in Subsection 5.1 of the main paper, our barycenters are the most accurate and exhibit substantially clearer and sharper structural details (see Figure 2 in the main paper), compared with results from entropic methods implemented in the python library "POT: Python Optimal Transport", while achieving the lowest computational cost among all benchmarks (See Table 1 in the main paper).


Below, we summarize the main concerns raised by the reviewers along with our responses. Detailed replies are provided in the individual responses to each reviewer.


| Questions              | Reviewer(s)          | Responses                                                                                                                             |
|------------------------|----------------------|----------------------------------------------------------------------------------------------------------------------------------------|
| higher dimensions?     | VvTD, Vdfa, h7bp     | We focus on **high-accuracy exact** barycenters for 2D or 3D distributions, as stated as a limitation. Existing high-dimensional methods rely on neural networks or regularization, *trading accuracy for scalability*. Even POT only supports entropic barycenters in 1D–2D. |
| more data example?     | DYXD, VvTD           | The handwritten-digits example was added in Appendix B3.                                                                               |
| adaptive gradient?     | VvTD                 | We added the adaptive Sobolev gradient (SGA-Adagrad) and evaluated it in Appendix B3.                                                  |
| irregular meshes?      | Vdfa                 | The algorithm can extend to irregular meshes using multigrid tools, at higher computational cost.                                      |
| compare WDHA?          | DYXD                 | Ours is a concave formulation; WDHA is nonconvex–concave. Our method enjoys global convergence, while WDHA does not (its projected variant is analyzed but projection is computationally infeasible). Our method is more stable. |
| other cost?            | h7bp                 | Our method applies to costs of the form \(c(x,y)=h(x-y)\), where \(h\) is strictly convex and even. The quadratic cost (Wasserstein barycenter) is a special case. |
| accuracy of OT maps?   | h7bp                 | A Gaussian example with relative error between estimated and true OT maps is reported in Appendix B4.                                  |
| optimal rate?          | DYXD                 | The convergence rate of SGA is optimal.                                                                                                |

---

### Meta-Review · Area_Chair_MGWa · 2026-01-06

**Summary:**

In their paper, the authors introduce Sobolev Gradient Ascent (SGA), a principled algorithm for computing the exact Wasserstein-2 barycenter of multiple input distributions discretized over a regular grid. The core contribution is a novel, constraint-free concave dual formulation of the barycenter problem, which eliminates the need for explicit $c$-concavity projection that still admits a simple and globally convergent optimization procedure. Building on this formulation, the authors propose a Sobolev gradient ascent method that is both theoretically grounded and practically efficient on low-dimensional grids. Overall, I think this is a solid paper.

(1) *Novelty and Technical Soundness*: All reviewers agree that the paper makes a meaningful theoretical contribution. The main novelty lies in the derivation of a constraint-free concave dual for the exact Wasserstein barycenter, which contrasts with prior approaches that either rely on entropic regularization or require expensive
$c$-concavity projection steps during optimization. This formulation allows the authors to apply Sobolev gradient ascent directly, leading to a clean convergence analysis with global guarantees. During the discussion period, the authors further strengthened the paper by clarifying the relationship to prior methods (especially WDHA), addressing concerns of Reviewer VvTD about stepsize selection by providing additional justification and adaptive variants, as well as refining claims in the related work to better reflect the positioning of the contribution.

(2) *Empirical Evaluation and Practical Impact*: The empirical evaluation demonstrates that SGA is competitive with, and often more stable than, existing exact barycenter computation methods in the literature. In response to reviewers' feedback, the authors added additional experiments, including handwritten digit barycenters and Gaussian settings with known ground truth. Moreover, they also evaluated the accuracy of the induced optimal transport maps, showing improved performance relative to prior methods, which strengthens the practical relevance of the approach.

The primary remaining concern raised by Reviewers VvTD, Vdfa and
h7bp is the restriction to low-dimensional, grid-based settings, which limits immediate applicability to very high-dimensional problems such as large-scale image benchmarks. However, the authors clearly acknowledge this limitation and appropriately frame the paper's goal as exact and theoretically principled barycenter computation, rather than scalable approximation in high dimensions. Within this scope, the contribution is non-trivial.

In summary, the rebuttal has addressed most of the concerns and strengthened the presentation, making this paper a strong candidate for acceptance, which paves the way for future work on extending exact barycenter methods or developing principled approximations beyond low-dimensional grids.

**Reviewer Concerns:**

Please refer to the summary.

**Reviewer Scores:**

Please refer to the summary.

---

### Decision · Program_Chairs · 2026-01-26

Accept (Poster)